# Chiral multi-curved shell metamaterials integrating compression-torsion and buckling mechanisms for ideal energy absorption

Chen-Xu Liu[1,2], Yizhi Zhang[1], Xinghao Wang[1], Gui-Lan Yu[3], Zhuo Zhuang[1] & Zhanli Liu [1] ✉

Metamaterials with compression-torsion or buckling mechanism have demonstrated significant potential for energy absorption. However, compression-torsion metamaterials easily trigger deformation, resulting in low load-bearing capacity, and buckling ones have high peak load with fluctuations, accompanying severe localized deformations. Here, we propose chiral multi-curved shell (CMCS) metamaterials that synergistically couple compression-torsion and buckling mechanisms, achieving high and smooth load curves. The compression-torsion mechanism enables metamaterials to convert compressive deformation into torsional deformation, preventing abrupt changes in local geometry. Simultaneously, the synergy of compression-torsion and curved shell ensures that the buckling provides high load-bearing capacity and avoids localized deformation. This coupled compression-torsion-buckling deformation enables the material to achieve high energy storage. Characterized by tests, the CMCS metamaterials exhibit enhanced energy absorption and tuneability. Compared with Kresling and hexagon metamaterials, the proposed design achieves a 20-fold higher specific energy absorption (SEA) and a 50% higher efficiency of energy absorption (EEA) owing to its higher and gentler plateau phase, respectively. Multiple drop tests demonstrate their reliable impact protection and reusability. CMCS metamaterials provide a novel concept for lightweight and high-strength protective structures or materials.

Metamaterials are artificial materials that exhibit non-conventional behavior not typically found in natural materials, determined by their microstructure rather than their composition[1–3]. Due to their special properties, such as compression-torsion[4], local buckling[5], auxeticity[6], wave invisibility[7], and lossless transmission[8], metamaterials have attracted significant attention across various fields[9–13]. Recently, metamaterials with high energy absorption capability and low density have garnered growing interest in impact and shock protection[14–18].

By combining lightweight characteristics with superior energy dissipation, energy-absorbing metamaterials achieve a balance of strength and mass, making them ideal for applications requiring efficient shock absorption with less material consumption[19–21]. Energy absorption in metamaterials is typically achieved through elastic or

[1]Applied Mechanics Lab., Department of Engineering Mechanics, School of Aerospace, Tsinghua University, Beijing, China. [2]Department of Mechanics, Beijing University of Technology, Beijing, China. [3]School of Civil Engineering, Beijing Jiaotong University, Beijing, China. ✉e-mail: liuzhanli@mail.tsinghua.edu.cn

plastic nonlinear deformation of their microstructures[15,22–25], such as honeycomb, lattice, and shell configurations[26–31]. Upon low- or medium-speed impact, these microstructures generally undergo bending, compressing, or stretching, thereby absorbing the impact energy[32–34]. Hence, by ingeniously designing microstructures, the energy absorption performance of metamaterials can be enhanced[35–37].

Based on the design of chiral microstructures, whose mirror configuration cannot be superimposed into themselves, metamaterials can exhibit compression-torsion property[38–40]. This feature holds great substantial promise for energy absorption, as rotational deformation under compressive loads helps dissipate energy, thereby reducing pressure on the protected structure[41–44]. The monotonic nature of torsional deformation results in a smooth stress plateau, bringing the energy absorption curve closer to the ideal shape[42]. However, the ease of deformation in compression-torsion metamaterials often leads to low load-bearing capacity[45–47], limiting their ability to fully absorb energy. While some compression-torsion metamaterials possess high load-bearing capacity, they require an additional force, applied perpendicular to the compression direction, to trigger the compression-torsion deformation[48–50], which is difficult to achieve in real impact or shock protection scenarios. Hence, to improve energy absorption, it is significant to preserve the compression-torsion property of metamaterials while simultaneously enhancing their load-bearing capacity.

Buckling is a common phenomenon in energy-absorbing metamaterials, inducing local material nonlinearity[51–53]. The pre-buckling state of microstructures provides these metamaterials with high load-bearing capacity[54]. However, buckling typically causes strain to concentrate in localized regions, altering its deformation pattern and resulting in a sharp force drop that deviates from ideal energy absorption behavior[55]. Although Snapp et al.[56] discovered a Palm microstructure characterized by a long and smooth plateau phase, its energy absorption relies on multilevel local buckling, which limits the efficient utilization of the material. This behavior is analogous to the energy absorption mechanism observed in foams[57] and is generally unfavorable for material reusability. If buckling is distributed across most areas, the material can be more fully utilized, and the deformation pattern may remain relatively uniform. This reduces the likelihood of a significant force drop and contributes to a near-ideal energy absorption curve. Recently, Fang et al.[58] proposed a chiral metamaterial composed of rods that achieved substantial recoverable elastic energy through twist buckling. Their study demonstrated that twisting induces deformation across the entire rod during buckling, thereby enhancing both load-bearing capacity and enthalpy. However, it did not specifically focus on how to achieve ideal energy absorption performance.

Here, we design a chiral multi-curved shell (CMCS) metamaterial for energy absorption, which integrates compression-torsion and buckling mechanisms. The synergy between compression-torsion behavior and curved shell geometry ensures sufficient material utilization and facilitates buckling-induced strain across most areas of the metamaterial, resulting in a smooth plateau phase and high load-bearing capability. Compared to a classical compression-torsion or buckling metamaterial with Kresling or hexagonal pattern, respectively, the proposed metamaterial demonstrates enhanced energy absorption. Using the laws observed in experimental data, we optimize a design with an improved energy absorption curve. Through multiple drop tests, it is confirmed that the metamaterial exhibits effective impact attenuation compared with the unprotected case and retains its performance upon reuse. The CMCS metamaterials provide a new design concept for energy-absorbing materials and structures in the field of impact and shock protection.

## Results

### Chiral multi-curved shell (CMCS) metamaterial with compression-torsion and buckling mechanisms

Figure 1a illustrates the microstructure of the proposed CMCS metamaterial. The inspiration of its compression-torsion mechanism comes from Kresling origami structures which guide this unique deformation mode through staggered spatial creases[59]. Kresling origami configurations are typically created by folding planar sheets, whereas the CMCS metamaterial can only be constructed using a fundamentally different approach. It is generated by sweeping and rotating a concentric polygon along the vertical axis (Supplementary Note 1), replacing the characteristic "two flat surfaces + one crease" of Kresling patterns with a continuous curved surface. This curved geometry facilitates controlled buckling, thereby enhancing the energy absorption capability. For avoiding the out-of-plane instability in rotation planes as depicted in Supplementary Note 2, two stiffeners, which are concentric polygonal plates with an in-plane width $w$ and a thickness $c$, are attached to the top and bottom of the main structure, thereby enhancing the vertical stiffness of rotation planes. The current structure is mirrored about the upper surface as the plane of symmetry. This resulting configuration ensures that only the middle plane rotates during compression[60], while the lower and upper planes do not undergo rotation, making it more suitable for impulse-driven impact or shock engineering. Furthermore, since the lower and upper planes are typically sealed in practical applications, multiple holes are drilled near these planes to prevent the incompressibility caused by the closed-cell effect.

Considering reusability and energy absorption capability, thermoplastic polyurethane (TPU) is chosen for metamaterials due to its excellent recovery property and relatively high modulus[61]. 3D printing technology with TPU is adopted to form the CMCS metamaterial. For comparison, a classical Kresling metamaterial and a regular hexagon metamaterial, each with same mass, plane polygon, and height to the CMCS metamaterial, are also printed, of which configurations are provided in Supplementary Note 3. Figure 1b presents the force-displacement curves through quasi-static compression experiments, referred to as energy absorption curves, for the three metamaterials, where the detailed information on experiments can be found in Supplementary Note 4. It is evident that the Kresling pattern exhibits a very low load-bearing capacity. Its specific energy absorption (SEA) is less than one-twentieth of that of the CMCS metamaterial. Although the hexagon pattern has a similar plateau force to the CMCS metamaterial, its peak force before densification is significantly higher—1.7 times the plateau force—resulting in notably low efficiency of energy absorption (EEA). Figure 1c shows their SEA and EEA, of which equations are given in the Methods section. Obviously, the CMCS metamaterial possesses a sufficiently high SEA and an EEA that significantly surpasses those of the other two metamaterials, aligning more closely with the ideal energy absorption curve described in the Methods section.

Figure 2a shows the stress nephograms of the three metamaterials at various displacements, along with their corresponding actual deformation states (detailed material properties and finite element analysis are provided in Supplementary Notes 5 and 6, respectively). Because the Kresling metamaterial consists of flat surfaces linked by hinge-like crease lines, stress is concentrated at the creases and remains relatively low. This limited stress amplitude prevents the TPU material from undergoing sufficient deformation, leading to very poor energy-absorption capability. Before buckling, the entire hexagon metamaterial is predominantly subjected to compressive stress, leading to a high load-bearing capacity. However, after buckling, the structure's geometry changes significantly, causing it to bend in a manner similar to that of a bent rod. Consequently, the load-bearing capacity of the hexagon metamaterial is reduced, resulting in a low EEA. It can be observed that the abrupt and significant change in structural configuration due to buckling is detrimental to EEA.

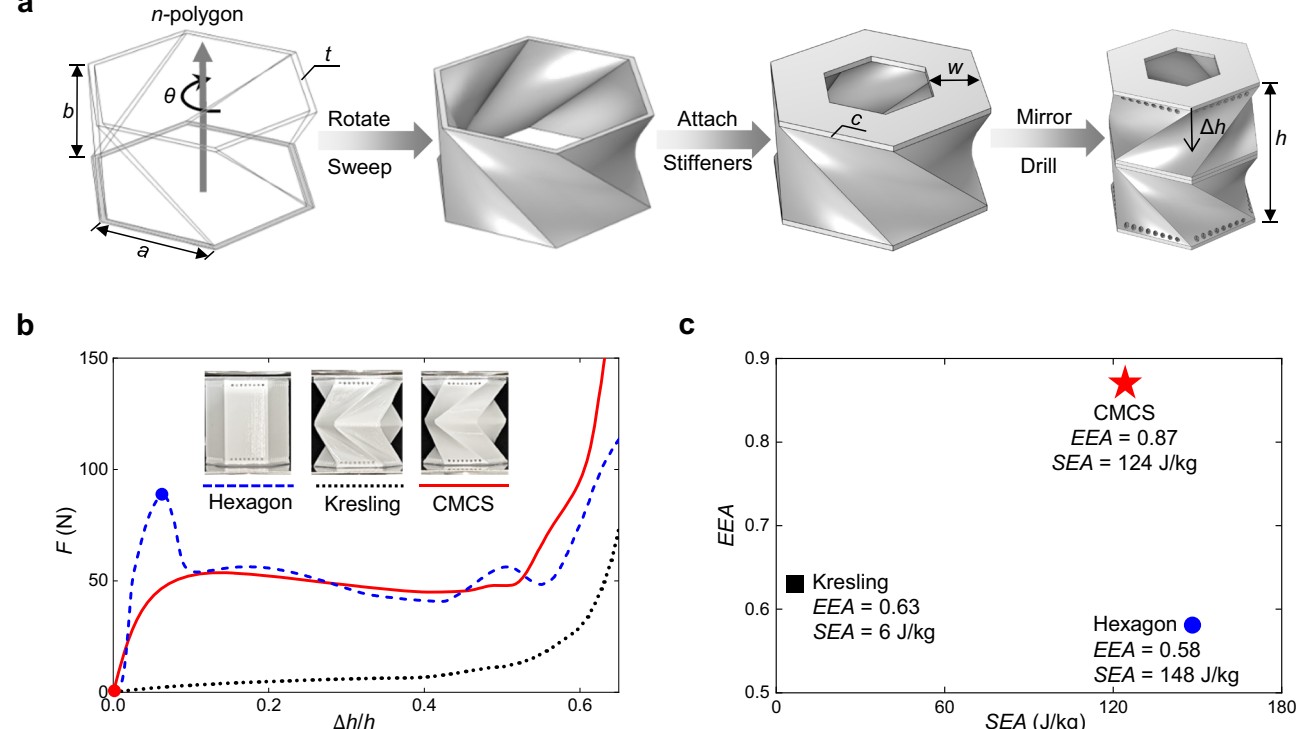

**Fig. 1 | The proposed CMCS metamaterial. a** The construction way of CMCS metamaterial; *n* is the side number of the polygon; *a* is the side length of the polygon; *b* is the sweep height of the polygon; *θ* is the rotation angle when the polygon sweeps; *t* is the in-plane thickness of the polygon; *c* is the thickness of stiffeners; *w* is the in-plane width of stiffeners; *h* is the height of the metamaterial and △*h* is the compression displacement. **b** Force-displacement curves of the hexagon, Kresling, and CMCS metamaterials by quasi-static compression experiments, where the blue dashed, black dotted, and red solid lines represent the hexagon, Kresling, and CMCS metamaterials, respectively; the blue point is the buckling position for the hexagon metamaterial and the red point is that for the CMCS metamaterial; the three metamaterials are designed with the same mass, and their actual 3D-printed masses are 7.6 g, 7.7 g, and 7.7 g, respectively; *F* is the force. **c** Specific energy absorption (SEA) and efficiency of energy absorption (EEA) of the three metamaterials, where the blue circle, black square, and red star points are SEA and EEA of the hexagon, Kresling, and CMCS metamaterials, respectively.

However, due to the compression-torsion mechanism, the configuration characteristics of the CMCS metamaterial remain largely unchanged throughout the deformation process. This consistency ensures a stable load capacity without abrupt fluctuations, resulting in a relatively smooth plateau phase.

Furthermore, as shown in Fig. 2b, the initially straight crease between two adjacent curved shells gradually becomes more curved as the compression displacement increases, indicating that the CMCS metamaterial enters a buckling state from the onset of compression. As shown in Fig. 2c, the curved shells of the CMCS metamaterial experience a compressing-rotating-buckling deformation mode under its synergy mechanism, ensuring that most of the material is effectively utilized for energy absorption. During the compression-torsion process shown in Supplementary Movie 1, all curved shells of the CMCS metamaterial undergo sufficient deformation owing to their buckling, enabling the overall metamaterial to achieve a high load-bearing capacity and absorb a significant amount of energy. Hence, owing to the unique compressing-rotating-buckling deformation mode, CMCS metamaterials exhibit a novel lightweight and high-strength impact-protection function.

Additionally, it can be inferred from Supplementary Movie 1 that the deformation mode of the CMCS metamaterial is primarily determined by the configuration of its microstructure, rather than by the periodicity. Therefore, the energy absorption performance of the overall metamaterial can be derived by analyzing its microstructure.

## Optimizing metamaterials for enhanced energy absorption performance

Twenty-six of compression experiments are conducted to investigate the relationship between characteristic parameters shown in Fig. 3a and energy absorption performance, with the aim of identifying an optimal metamaterial. Figure 3 shows the energy absorption curves of CMCS metamaterials with varying *t*, *n* and *θ*, with detailed configuration information provided in Supplementary Note 7. The variation in EEA and SEA with respect to thickness *t* is shown in Fig. 4a. Obviously, *t* has little effect on EEA, while the value of SEA increases significantly as *t* increases. The parameter *t* does not significantly alter the overall configuration characteristics of the metamaterial, leading to minimal changes in the shape of the energy absorption curve as shown in Fig. 3d. Consequently, the value of EEA remains relatively stable. Furthermore, increasing *t* leads to a thicker curved shell, enabling each unit of material to absorb more energy under the same deformation, thereby enhancing the SEA. This phenomenon is consistent with the behavior described in Supplementary Note 8. Both *n* and *θ* significantly influence the overall configuration of the CMCS metamaterial, as shown in Fig. 3b–c, resulting in substantial changes in the energy absorption curve, as illustrated in Fig. 3e–f. Therefore, these two parameters greatly affect EEA and SEA simultaneously, as shown in Fig. 4b–c. These results highlight the programmability of CMCS metamaterials: the plateau force can be tuned to meet different protection requirements simply by adjusting the thickness *t*, while the overall shape of the energy absorption curve remains largely unchanged; in addition, by varying the parameters *n* and *θ*, the positive

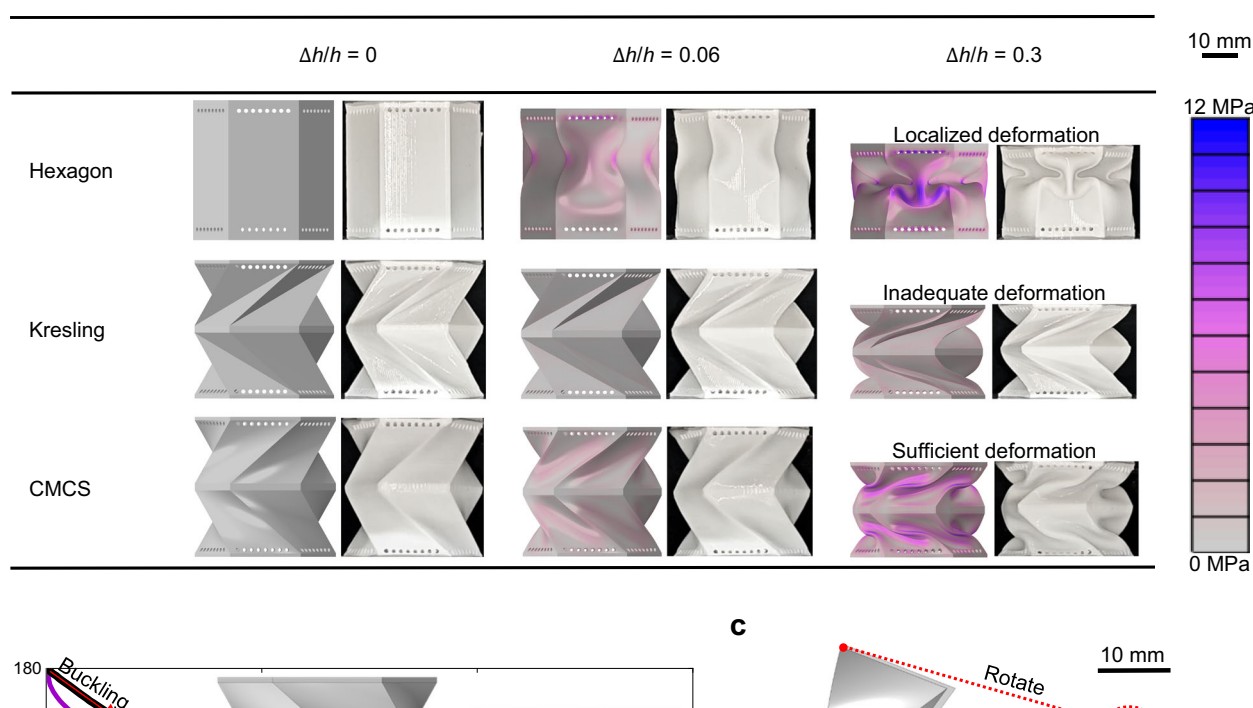

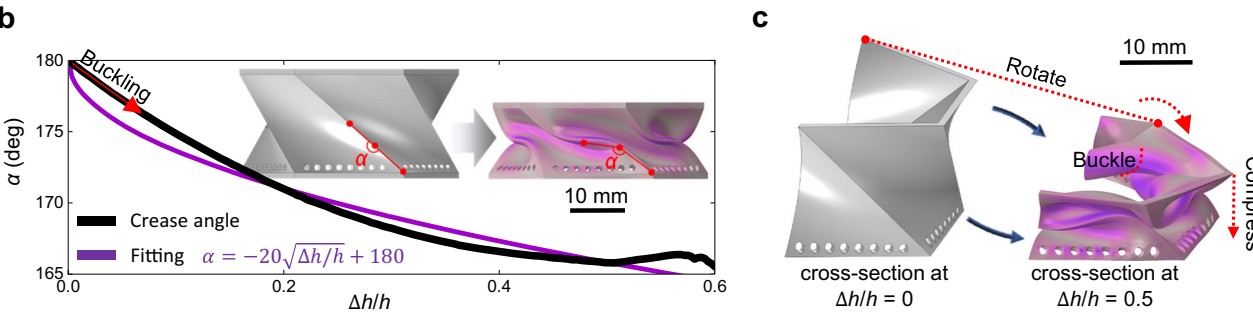

**Fig. 2 | Compression-torsion and buckling mechanisms of the CMCS metamaterial. a** Mises stress nephograms and corresponding deformations of the hexagon, Kresling, and CMCS metamaterials, where $h$ is the height of the metamaterial and $\Delta h$ is the compression displacement. **b** Buckling characteristic curve, that the angle ($\alpha$) between the line from the 1/4 point to the bottom point and the line from the 1/4 point to the 1/2 point in a crease varies with the compression displacement. **c** Sectional view of Mises stress nephogram for the main structure of the CMCS metamaterial at $\Delta h/h = 0.5$.

and negative stiffness characteristics of the energy absorption curve can also be tailored. Further discussion on the programmability of CMCS metamaterials through parallel connection and matryoshka-like designs is provided in Supplementary Note 9.

It can be seen from Fig. 4b that the CMCS metamaterial with $n = 6$ has the highest EEA than those with other $n$. Furthermore, when $\theta$ is 72°, it achieves a higher EEA compared to others, as shown in Fig. 4c. Therefore, the CMCS metamaterial with $n = 6$ and $\theta = 72°$ exhibits a notable high EEA. An ideal energy absorption curve requires an appropriate plateau force as described in the Methods section. Therefore, for CMCS metamaterials, a higher SEA is not inherently advantageous; rather, a SEA that aligns with an appropriate plateau force is desirable. From energy absorption curves in Fig. 3d–f and their corresponding SEA values, it can be inferred that a higher SEA often corresponds to a higher plateau force. Based on the laws observed in Figs. 3d and 4a, the SEA and the plateau force of the metamaterial with $\theta = 72°$ can be easily adjusted by varying $t$ while maintaining a similar EEA.

To conservatively prevent the instability of the middle plane described in Supplementary Note 2, $w$ of the stiffener is set to be a relatively high value, which results in unnecessary material consumption. Hence, Fig. 4d discusses how reducing excess material by varying $w$ affects the energy absorption performance of the metamaterial with $n = 6$, $\theta = 72°$, $t = 1$ mm, $h = 40$ mm, $a = 20$ mm, $b = 18$ mm, and $c =$

1 mm. It can be observed that the plateau phase of the blue dashed-dotted line exhibits a visible decrease, which results in a reduction in EEA. Supplementary Note 10 illustrates the EEA and SEA values for these CMCS metamaterials, revealing that the metamaterial with $w = 3\sqrt{3}$ mm absorbs more energy per unit mass while maintaining a comparable EEA of 0.91. Its SEA reaches 75.7 J/kg, representing a 16% improvement over the conservative configuration. Hence, the CMCS metamaterial shown in Fig. 4e is a better choice.

Furthermore, Fig. 4f–g compare the EEA and SEA of CMCS metamaterials with those of other existing energy-absorbing metamaterials fabricated with TPU[50,56,62–68]. Compared to the highest EEA of 0.83 achieved by other metamaterials, CMCS metamaterials exhibit an improvement of 0.08. This incremental gain represents a meaningful step toward the ideal value of 1. Furthermore, CMCS metamaterials outperform other metamaterials in SEA, combining lightweight design with high energy absorption. Hence, CMCS metamaterials demonstrate meaningful advancements in EEA and SEA.

In addition, CMCS metamaterials can be extended into 3D lattices through periodic arrangement while maintaining reliable energy absorption performance, as discussed in Supplementary Note 12.

## Evaluating CMCS metamaterials
To evaluate the energy absorption performance of the optimized CMCS metamaterial, a periodic structure consisting of $3 \times 3 \times 1$ units is

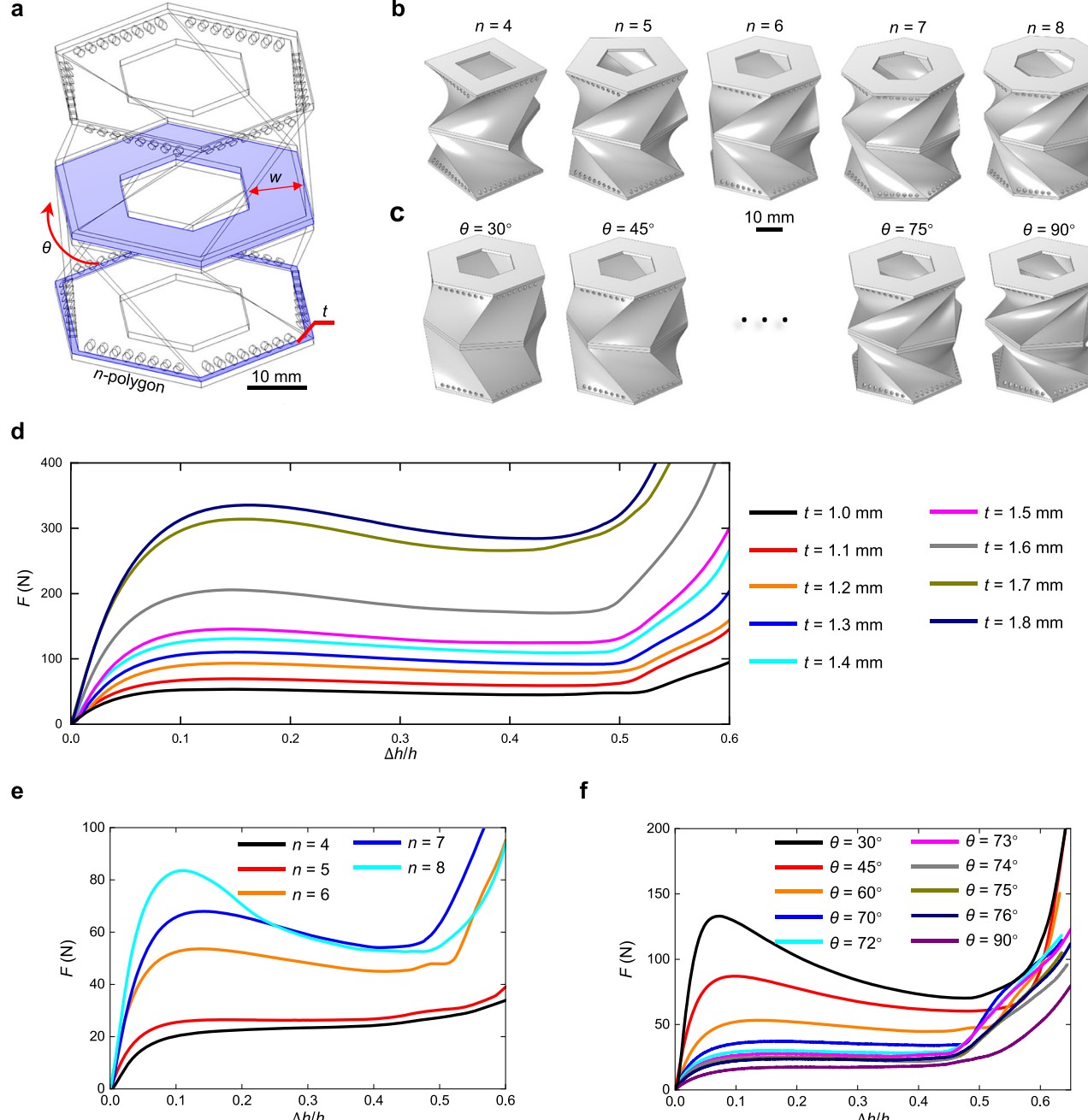

**Fig. 3 | Influence of geometric parameters on the force-displacement curves of CMCS metamaterials. a** Considered variable parameters including $t$, $n$, $\theta$, and $w$, which are the in-plane thickness of the polygon, side number of the polygon, rotation angle when the polygon sweeps, and in-plane width of stiffeners, respectively. **b** Models with varying $n$ for 3D-printing. **c** Models with varying $\theta$ for 3D-printing. **d** Experimental force-displacement curves under quasi-static compression when $t$ varies between 1 mm and 1.8 mm, where $F$ is the force. **e** Those when $n$ varies between 4 and 8. **f** Those when $\theta$ varies between 30° and 90°. Only one parameter varies while the others remain constant with the following values: $t = 1$ mm, $\theta = 60°$, $n = 6$, $h = 40$ mm, $a = 20$ mm, $b = 18$ mm, $c = 1$ mm, and $w = 5\sqrt{3}$ mm.

printed, as shown in Fig. 5a. It is then compared with a classic re-entrant metamaterial renowned for its strong energy absorption capability[69], as illustrated in Fig. 5b. To ensure a fair comparison, the thickness is adjusted so that both metamaterials have a similar plateau force per unit area, allowing each to meet the same protection requirements. Detailed geometric information on them can be found in Supplementary Note 13. The energy absorption curves for the two metamaterials under quasi-static compression are compared, where the force is divided by the area to offset area differences. The corresponding deformations are shown in Supplementary Movies 2 and 3. Obviously, the CMCS metamaterial exhibits a smoother curve, as its deformation pattern remains consistent throughout the compression.

Furthermore, Fig. 5c compares the EEA, SEA, plateau phase, densification displacement, undulation of load-carrying (ULC), and relative density for both the CMCS and re-entrant metamaterials. Among these energy absorption metrics, of which values are provided in Supplementary Note 13, the CMCS metamaterial surpasses its re-entrant counterpart, particularly in terms of SEA, which is 312.5% of the latter. Meanwhile, the former has only about 41% of the relative density of the latter, highlighting the lightweight advantage of CMCS metamaterials. Two other re-entrant metamaterials with different microstructures are also tested, and their energy absorption performance remains significantly inferior to that of the CMCS metamaterial, as illustrated in Supplementary Note 13. In addition, at the same relative density of

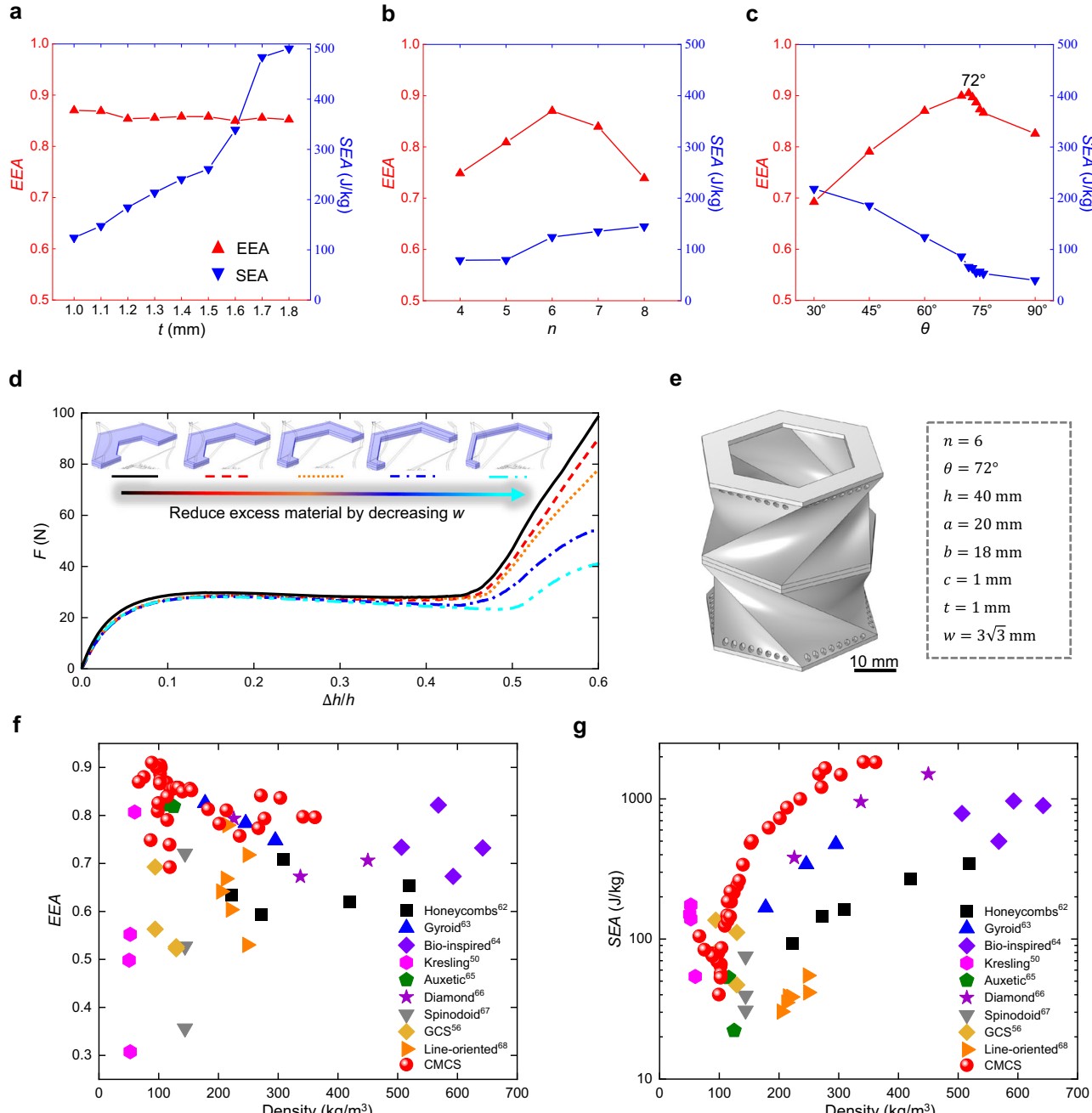

**Fig. 4 | Optimizing CMCS metamaterials to closely approximate an ideal energy absorption curve. a** The effect of $t$ on EEA and SEA of force-displacement curves in Fig. 3d, where $t$ is the in-plane thickness of the polygon. **b** The effect of $n$ on EEA and SEA of force-displacement curves in Fig. 3e, where $n$ is the side number of the polygon. **c** The effect of $\theta$ on EEA and SEA of force-displacement curves in Fig. 3f, where $\theta$ is the rotation angle when the polygon sweeps. **d** Reducing excess material by varying $w$ affects the energy absorption performance of the CMCS metamaterial with $t = 1$ mm, $n = 6$, $\theta = 72°$, $h = 40$ mm, $a = 20$ mm, $b = 18$ mm, and $c = 1$ mm, where $w$ is the in-plane width of stiffeners and its values for the black,

red, orange, blue, and cyan lines are $5\sqrt{3}$, $4\sqrt{3}$, $3\sqrt{3}$, $2\sqrt{3}$, and $\sqrt{3}$ mm, respectively. **e** The optimized CMCS metamaterial. **f** Ashby plot comparing the EEA of CMCS metamaterials with that of other metamaterials fabricated with TPU. **g** Ashby plot comparing the SEA of CMCS metamaterials with that of other metamaterials fabricated with TPU. All EEA and SEA values in the Ashby plots except those of CMCS metamaterials are sourced from existing studies[50,56,62–68]. Detailed information on additional experimental points of CMCS metamaterials in **f**, **g** can be found in Supplementary Note 11.

28.7%, as shown in Supplementary Fig. 14, the SEA of the CMCS metamaterial is up to 16.4 times that of the re-entrant metamaterial, representing a qualitative leap. It is evident that the CMCS metamaterial exhibits an improved energy absorption curve and enhanced energy absorption performance.

It is worth noting that energy absorption curves obtained from quasi-static compression tests are generally considered reliable proxies for evaluating low- and medium-speed impact resistance[70–72]. This is

because they capture the dominant energy-absorption deformations, such as bending, compressing, and stretching, that are normally rate-insensitive[73–75]. Therefore, the quasi-static curve provides a sound basis for assessing the energy absorption of metamaterials under low- to medium-speed impact loading conditions. Multiple drop tests are conducted to demonstrate the impact protection performance and reusability of CMCS metamaterials. The CMCS metamaterial is attached to the hammer, as shown in Fig. 5d, simulating practical

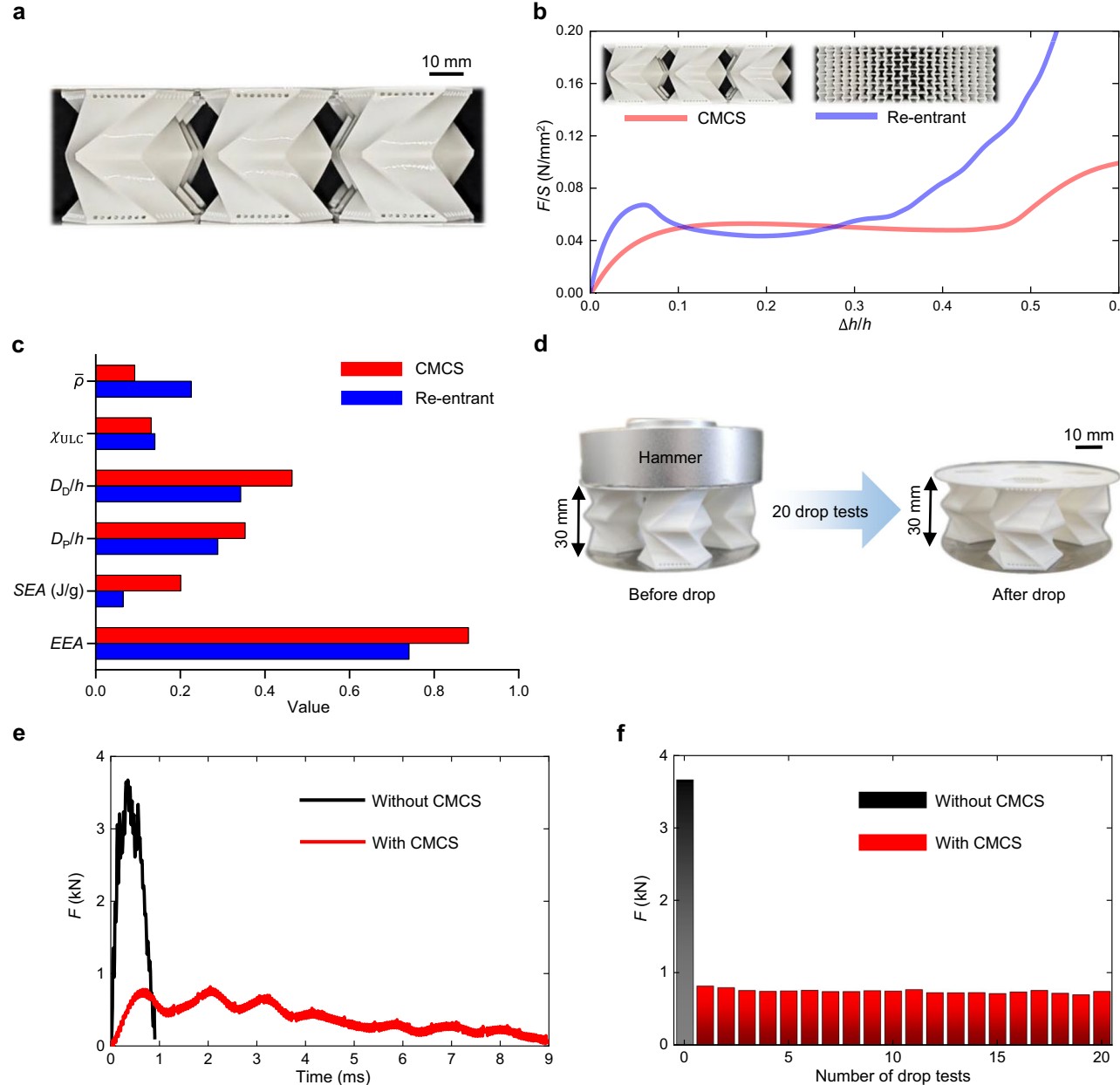

**Fig. 5 | Energy absorption performance evaluation. a** 3D-printed CMCS metamaterial with 3 × 3 × 1 units. **b** Force-displacement curves of the CMCS (red line) and re-entrant (blue line) metamaterials by quasi-static compression experiments, where $F$ and $S$ are the force and pressure area, respectively. **c** Comparing CMCS (red bar) and re-entrant (blue bar) metamaterials in EEA, SEA, plateau phase ($D_P$), densification displacement ($D_D$), undulation of load-carrying ($\chi_{ULC}$), and relative density ($\bar{\rho}$) of which equations and values are provided in the Methods section and Supplementary Note 13, respectively, where $h$ is the height. **d** The CMCS metamaterial attached to the hammer, where the hammer mass is 1 kg and th**e** drop height is 1 m. **e** Force-time curves with (the red line) or without (the black line) the CMCS metamaterial. **f** Twenty drop tests are conducted with the same CMCS metamaterial and one without it; each red bar represents the peak force measured in the tests with the CMCS metamaterial, while the black bar represents the peak force without it.

protective drop scenarios. Considering the size limitation of the hole through which the hammer passes as shown in Supplementary Fig. 15, the main dimensions of the CMCS microstructure is proportionally reduced by 25%, and a 2 × 2 arrangement is adopted. Detailed information regarding the drop hammer machine and the CMCS metamaterial is provided in Supplementary Note 14. As illustrated in Fig. 5e, the peak force experienced with the CMCS metamaterial during the first drop is reduced to 22.3% of the peak force observed without it. Even after 20 repeated drops, the attenuation effect of the CMCS metamaterial remains nearly unchanged from its initial value, as depicted in Fig. 5f. These results confirm the enhanced impact protection performance and reusability of CMCS metamaterials.

In conclusion, we propose a construction way for CMCS metamaterials with improved energy absorption performance by integrating the compression-torsion and buckling mechanisms. The compression-torsion mechanism converts compressive deformation into torsional deformation, and the synergy of compression-torsion and curved shell ensures buckling-induced strain across most areas, sufficiently utilizing the material for energy absorption. This coupling effect endows CMCS metamaterials with smooth plateau phases and high load-bearing capacity. Compared with the compression-torsion Kresling origami metamaterial, the CMCS metamaterial exhibits a 20-fold increase in SEA. Relative to the hexagon metamaterial, it achieves a 50% increase in EEA. Moreover, it exhibits superior energy absorption

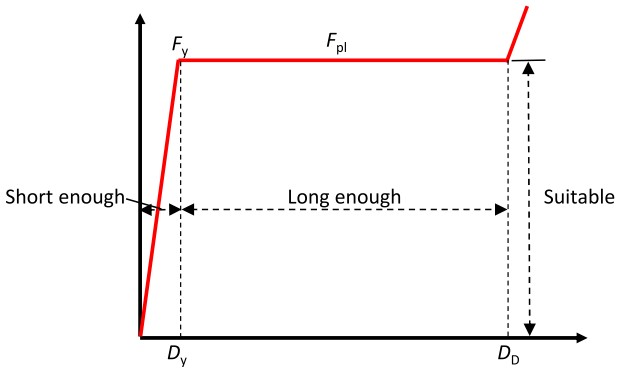

**Fig. 6 | Ideal energy absorption curve.** $F_y$ is the yield force; $F_{pl}$ is the plateau force; $D_y$ is the yield displacement; $D_D$ is the densification displacement.

metrics compared to a classic re-entrant metamaterial, particularly with its SEA reaching 312.5% of that of the latter. Additionally, multiple drop tests also confirm its enhanced impact protection and reusability. Overall, the integration design of compression-torsion and buckling mechanisms endows energy-absorbing metamaterials with high SEA and EEA, paving a new way for protection engineering applications, such as buildings, armored vehicles, safety helmets, and drones.

## Methods
### 3D printing of metamaterials
All metamaterials and microstructures in this study are fabricated using fused deposition modeling (FMD), a type of 3D printing technology, with TPU as the material. The TPU filaments are sourced from Bambu Lab, which are of the type TPU 95 A HF. 3D solid models are generated and exported as STEP files. These files are then imported into Bambu Studio and sliced with a layer thickness of 0.16 mm for the quasi-static compression samples and 0.08 mm for the drop test sample which has a smaller size requiring higher printing resolution, respectively. The printing temperature is set to be 230 °C, and the bed temperature is set to be 35 °C. The printing speed ranges from a maximum of 30 mm/s to a minimum of 1 mm/s, with slower speeds employed as the inclination decreases.

### Quasi-static compression tests
Quasi-static compression tests on 3D-printed metamaterials are conducted using a Universal Testing Machine (UTM) at room temperature, with a crosshead speed of 0.4 mm/s. The upper and lower planes of metamaterials are bonded to a $50 \times 50 \times 0.6$ mm steel plate using AB glue.

### Important metrics for evaluating energy absorption performance
Densification displacement, plateau phase, undulation of load-carrying (ULC), efficiency of energy absorption (EEA), and specific energy absorption (SEA) are important metrics for evaluating the energy absorption performance of metamaterials.

Densification displacement refers to the displacement from the initial undeformed state to the compacted state during compression. In an energy absorption curve, it can be identified as the displacement corresponding to the maximum value of the energy-absorbing efficiency function[56,76], which is as follows:

$$D_D = \text{argmax}\, \frac{1}{F(D_a)D_H}\int_0^{D_a} F(D)\,\mathrm{d}D \qquad (1)$$

where $D$ is the displacement and $D_H$ is the initial height of the compressed structure or material; $F(D_a)$ refers to the force at $D = D_a$; $D_D$ is the densification displacement. A larger densification displacement $D_D$

typically makes the normal working range longer, enabling greater energy absorption.

The plateau phase refers to the phase of deformation in which a relatively constant load is maintained over a longer deformation range. It is generally the distance from the yield displacement to the densification displacement, which can be expressed as follows:

$$D_P = D_D - D_y \qquad (2)$$

where $D_y$ is the yield displacement and $D_P$ is the plateau phase. A longer plateau displacement $D_P$ allows for extended energy absorption before densification occurs.

ULC can quantify the relative fluctuation of force in the compression process, which can be calculated as[32]:

$$\chi_{ULC} = \frac{\int_0^{D_D} |F(D) - F_{avg}|\,\mathrm{d}D}{\int_0^{D_D} F(D)\,\mathrm{d}D} \qquad (3)$$

where $F_{avg}$ is the average force from the beginning to the densification displacement. A smaller ULC $\chi_{ULC}$ generally results in a more gradual variation of the plateau force.

EEA is defined as the ratio of the energy absorbed before densification displacement to the product of the peak force prior to compaction and the densification displacement. It can be calculated by the following equation[32]:

$$EEA = \frac{1}{F_{peak}D_D}\int_0^{D_D} F(D)\,\mathrm{d}D \qquad (4)$$

where $F_{peak}$ is the peak force from the beginning to the densification displacement. A higher EEA implies that more energy can be absorbed before reaching the peak or damage threshold.

SEA can reflect the amount of energy absorbed per unit mass. In this study, it is calculated when $D = D_D$, and can be written as follows[32]:

$$SEA = \frac{\int_0^{D_D} F(D)\,\mathrm{d}D}{m} \qquad (5)$$

where $m$ is the mass. An increased SEA corresponds to more energy absorbed per unit mass, reflecting enhanced material utilization.

### Ideal energy absorption curve
Figure 6 shows the schematic diagram of an ideal energy absorption curve, and the standards are as follows: (a) EEA approaches 1; to achieve this goal, ULC must be close to 0, the plateau force should be equal to the yield force, and the displacement interval of the linear stage is as small as possible, which normally refers to $D_y$; (b) the plateau phase $D_P$, namely $D_D - D_y$ is long enough; (c) The plateau force is close to the damage threshold of a protected structure.

### Drop tests
Drop tests are conducted to assess the impact protection and reusability of CMCS metamaterials under dynamic loading conditions. The specimen is impacted using a custom-built drop-weight system featuring a guided vertical track and a cylindrical steel hammer (the mass is 1 kg, the diameter is 80 mm, and the drop height is 1.0 m). The CMCS metamaterial is attached to the hammer head to simulate practical protective scenarios, such as those encountered in sports helmets. Considering that the material typically experiences only a limited number of impacts before replacement, as in helmet use, 20 impact cycles at the same drop height are performed to balance multi-impact protection assessment with experimental costs and to evaluate its reusability.

## Finite element analysis

The hyperelastic Yeoh model is used to describe the nonlinear behavior of TPU, of which equation is as follows:

$$W = C_1(I_1 - 3) + C_2(I_1 - 3)^2 + C_3(I_1 - 3)^3 \qquad (6)$$

where $W$ is the strain energy density; $C_1$, $C_2$, and $C_3$ are the material coefficients; $I_1$ is the first invariant. The material coefficients $C_1$, $C_2$, and $C_3$ are determined by fitting the experimental stress-strain curve, resulting in values of 5.1 MPa, −2.5 MPa, and 0.8 MPa, respectively.

The commercially available software package ABAQUS 2022 is adopted to model the deformations of metamaterials printed with TPU. A fixed boundary condition is applied to the lower surface of a metamaterial, while a displacement load is applied to its upper surface to simulate the quasi-static compression. Eight-node linear hexahedral elements with reduced integration (element type C3D8R) and ten-node modified quadratic tetrahedral elements (element type C3D10M) are used to mesh metamaterials.

## Data availability

The force-displacement curves, metrics for evaluating energy absorption performance, buckling characteristic curve date generated, and STEP files used for 3D-printing in this study are provided in the Supplementary Information and the public repository at https://doi.org/10.6084/m9.figshare.28777931. Additional details on data that support the findings of this study will be made available by the corresponding author upon request.

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

## Acknowledgements

The authors acknowledge the financial support by the National Natural Science Foundation of China, No. 12402103 (C.-X.L.), 12525208 (Z.L.), and 11972205(Z.L.), the National Key R&D Program of China, No. 2022YFC2402700 (Z.L.), China Postdoctoral Science Foundation, No. 2024M751634 (C.-X.L.), and Postdoctoral Fellowship Program of CPSF, No. GZC20231310 (C.-X.L.).

## Author contributions

C.-X.L. conceived the idea. C.-X.L. and Z.L. designed the study. C.-X.L., Y.Z. and X.W. conducted experiments and finite element simulations. C.-X.L. analyzed data, interpreted the results, and drafted the manuscript. Y.Z., G.-L.Y. and Z.Z. participated in discussions and contributed to data interpretation. Z.L. and C.-X.L. secured funding. Z.L. supervised the project and provided critical revisions. All authors reviewed, discussed, and approved the final manuscript.

## Competing interests

The authors declare no competing interests.
