## [Transparent Peer Review file · Nature Communications]

Chiral multi-curved shell metamaterials integrating compression-torsion and buckling mechanisms for ideal energy absorption

Corresponding Author: Professor Zhanli Liu

Version 0:

Reviewer comments:

Reviewer #1

(Remarks to the Author)

This work proposes a chiral multi-curved shell (CMCS) metamaterial that utilizes the folding mechanics of shells to achieve enhanced energy absorption and recovery properties. The general concept of the work is interesting, but substantial improvements are necessary before it can be considered for publication. The main concerns from the reviewer are as follows:

- (1) The proposed design closely resembles existing origami metamaterials. The authors need to clearly justify how their design differs from these existing structures to demonstrate the novelty of the study.
- (2) The design presented is essentially an integration of multiple one-dimensional CMCS cylinders into two-dimensional panel-like functional components. However, the current version of the manuscript does not include any results demonstrating the potential for extending this structure into three-dimensional lattices or more complex functional systems. This requires significant justification.
- (3) A critical aspect of metamaterials is their programmability. The paper lacks sufficient discussion on the programmability of the proposed design and its mechanical and functional performance. Additional results and a more systematic investigation are needed; otherwise, the design's applicability remains highly limited.
- (4) There is extensive research on energy-absorbing metamaterials, but the literature review in this work is insufficient. Numerous studies have used strut-based or shell-based structures to engineer nonlinear behaviors in lattices, and these should be appropriately reviewed, including but not limited to:
Moestopo, W. P., Mateos, A. J., Fuller, R. M., Greer, J. R., & Portela, C. M. (2020). Pushing and pulling on ropes: hierarchical woven materials. *Advanced Science*, 7(20), 2001271.
Gao, Z., Ren, P., Wang, H., Tang, Z., Wu, Y., & Wang, H. (2024). Additive manufacture of ultrasoft bioinspired metamaterials. *International Journal of Machine Tools and Manufacture*, 195, 104101.
Wang, P., Yang, F., Zheng, B., Li, P., Wang, R., Li, Y., ... & Li, X. (2023). Breaking the tradeoffs between different mechanical properties in bioinspired hierarchical lattice metamaterials. *Advanced Functional Materials*, 33(45), 2305978.
- (5) Is the proposed strategy confined to hexagonal channels? If so, the generality of the design is limited and must be significantly improved.
- (6) The reported properties lack systematic comparison with existing structures in the literature. Although Figure 1c provides some comparisons, they are limited and based solely on author-defined cases. A more comprehensive comparison, such as Ashby charts with a broader set of topologies, is essential to evaluate the true impact of this work.
- (7) The authors must be very careful in discussing energy absorptions, as it can differ significantly across quasi-static, dynamic, and impact.

Reviewer #2

(Remarks to the Author)

The paper presents an intuitive, yet novel and noteworthy, design concept for energy absorbing mechanical metamaterials, called chiral multi-curved shells, that should be of interest to the applied mechanics community. Numerical and experimental data support the hypothesis that the design efficiently transfers axial loads to the distributed deformation of the metamaterial

walls, thereby resisting compression of the entire body without localizing strain. This result is contrasted with the behavior of hexagon and Kresling metamaterials which, respectively, excessively localize strain and poorly transfer axial loads to wall deformations. A parametric study is performed to understand the role of the design architecture and optimize the structure according to metrics for energy absorption. The capabilities for applications in impact mitigation are then discussed. Sufficient details are provided to enable reproduction of the fabrication, simulation, testing, and data analysis presented.

The results appear significant and valuable, and the conclusions presented the "Chiral multi-curved shell (CMCS) metamaterials with compression-torsion and buckling mechanisms" and "Optimizing metamaterials for an excellent energy absorption curve" sections are logically sound with sufficient evidence. However, I have three major concerns that should be addressed with revision:

(1) Insufficient explanation of metrics: The metrics used for evaluation of the metamaterials (specific energy absorption, energy absorption efficiency, plateau phase, densification displacement, and undulation of load-carrying) are defined in Supplementary Note 5, rather than the main text or the methods, and are used without clarification of their significance. The text frequently refers to an ideal energy absorption behavior, which is presented in Supplementary Note 6, but this concept does not appear to be clarified in supporting citations 41 or 52. These details should be elucidated in the main text for the paper to reach the general audience of the journal rather than the audience of a technical journal devoted to energy absorption. Without further explanation or expertise, it is not possible to interpret the significance of the results and to accept the conclusions of the paper.

(2) Unfair comparison of designs: In contrast with the thorough comparison of the CMCS to the hexagon and Kresling metamaterial, it is not evident the comparison of the CMCS to the re-entrant metamaterial is appropriately controlled. While the results presented are normalized with respect to the pressure area and the plateau force per unit area is matched via tuning of the thicknesses, there are multiple variables in the re-entrant metamaterial, particularly the number of cells, which are not discussed but may influence the results such as the specific energy absorption that depends on the total mass of the system. Furthermore, the pressure areas reported in Supplementary Note 12 should be properly motivated, specifically that of the CMCS metamaterial for which each tube has an open end but the reported pressure area is three times larger than that of the re-entrant metamaterial. Thus, the logic of the comparison used to reach the conclusion "It is evident that the CMCS metamaterial exhibits an excellent energy absorption curve and outstanding energy absorption performance" should be supported with additional evidence or logical arguments.

(3) Missing connection from statics to impact: The drop tests compare force measurements in the presence of a CMCS and with no energy absorber. While this provides sufficient evidence to support the claims regarding the reusability of the metamaterial as fabricated with TPU, the claim "These results confirm the excellent impact protection performance..." requires additional evidence or logical arguments directly connecting the quasi-static testing presented in the rest of the paper to the dynamic drop-test for impact testing. Furthermore, any implications of changing the printed layer thickness from 0.16 mm to 0.08 mm and from a 3x3 arrangement (Fig 3a) to an apparent 2x2 arrangement (Fig 3d) should be clarified. This connection is crucial to reach the statements made in "Potential applications" which are all impact related.

Reviewer #3

(Remarks to the Author)

Seems like the adjectives used in the first two paragraphs are a little bit superlative, or could be somehow exaggerated (extraordinary, exceptional, impressive).

Same in line 65, the use of outstanding seems too much.

As a reference, this seems more appropriate: significantly improved energy absorption

Line 70, eliminate word extensive,

line 72, change word outstanding, and also if you are going to claim that, compared to what, and how does this perform with other research projects?

line 73, novel may not be again a proper selection of adjectives, many other metamaterials shown in other published works show good energy absorption properties, so it is not really a new approach.

Intro or a subsequent section must include a more comprehensive research literature analysis.

Subsequent section, before results and conclusion, should be considered to include information regarding theoretical background, materials and methods. After finishing, it was noted that there is a methods section at the end, and supplementary information, which helped us understand a little bit better the design phase. It is suggested that some material should be included to understand the whole approach in a better way.

Difficult to follow through the text and with figures, when too many ideas are presented in one figure (applies for fig 1 and 2), even some sections required to move to supplementary notes, which do not help to have a proper flow of information

Good explanation for the reasoning behind the design of CMCS, from a Kresling design.

Line 93, mentions that comparison samples have similar mass, and feature parameters, how close they are in terms of mass, relative density, that can allow to guarantee a fair comparison?

Line 143, check use of excellent,

Line 144, dozens? Meaning? Could it mean 20 something, or 200 something? Figure 2 shows several parameters, but it is not clear if all the combinations were tested, a summary table of the parameters, combinations, replicas should be included. It is important to mention how many replicas and if a design of experiments methodology was used to guarantee statistical confidence. Also, it is not clear how all these changes of parameters result in significant differences in mass, relative density, that may mislead the interpretation of results.

Good comparison in the subsection with respect to the reentrant.

Section of potential applications should be moved to conclusions, no need for figures, unless they were tested somehow.

Too many repetitions of superlative adjectives, line 230, outstanding.

As a summary, it is a good work, needs some polishing in terms of format and structure, tests are good, but need some more details on the structure of the experiments, and more supporting figures, and detailed analysis in text.

Version 1:

Reviewer comments:

Reviewer #1

(Remarks to the Author)

The review thanks the authors for the revision. Most of the concern has been addressed. however, critical points as follows need to be addressed before publication:

(1) The explanation of the difference between the proposed work and existing studies is not convincing enough. Currently, it is more like a technical explanation, where all structures can have technical differences in stress distributions and mechanisms. What we are more interested is the new functionalities, properties, performance, and unique observations given, which are still absent in current submission.

(2) The response "Based on our experiments and numerical simulations, it is challenging to extend CMCS cylinders into three-dimensional (3D) lattices while preserving the deformation mode of individual cells" is not acceptable. In fact, this can raise significant challenges since the proposed structures are shown to have limitations in 3D cases.

(3) The results of the Ashby charts only show moderate advantages in SSA and SEA.

Reviewer #2

(Remarks to the Author)

The initial review requested (1) elaboration of metrics used in analysis, (2) confirmation designs are fairly compared, and (3) more rigorous connection between static and impact analysis. The following revision mostly address the requests and significantly enhance the paper, but there remains one concern regarding the control used in Point (2):

(1) The metrics are clearly explained using both mathematical formulae and plain language in the Methods section. Furthermore, the plain language explanation of the metrics enable a broader audience to appreciate the physical significance of the ideal energy absorption curve and the consequences for deviations from the ideal curve.

(2) Additional re-entrant architectures with a variable number of cells and wall thicknesses, but fixed mass density, are tested. Ashby plots that showcase the scaling of the novel architecture in comparison to existing architectures are constructed.

(3) Satisfactory references are provided to rigorously connect the static response to the impact response.

The superiority of the CMCS to the re-entrant metamaterial is primarily quantified according to the SEA metric which depends on the force integrated up to the densification displacement and the total mass. Therefore, the conclusion that the CMCS is generally superior to the re-entrant metamaterial, rather than for specific cases, requires showing that the mass of the re-entrant metamaterial increases faster than the integrated force as features such as the number of cells or wall thickness are changed. The scaling of the re-entrant metamaterial SEA versus mass density on the Ashby plot (which may currently be labeled as auxetic?) would resolve this concern once and for all. If this follows directly from results that are reported in the existing paper, then it should be clarified.

Reviewer #3

(Remarks to the Author)

Line 25 programmability, probably is better to use tuneability

Also from line 25

Compared to 25 traditional compression-torsion or buckling ones, the metamaterial achieves a 20-fold increase in 26 specific energy absorption (SEA) through a higher plateau force or a 50% increase in efficiency 27 of energy absorption (EEA) due to a gentler plateau phase, respectively.

Compared to what, the metamaterial has either feature? Kind of confusing the sentence.

Line 32, exotic, change for non-conventional behavior or something like that..

Line 44, recommended to introduce or explain what is a chiral microstructure.

Line 67, careful with statements that make claims of better performance without reference of how the comparison is made.

How much is significant improved? Better statement in line 70 of a clear comparison.

Again in line 74, a claim without much support and using a kind of ambiguous adjective (reliable)

For the experiments shown in fig, does the reentrant and the CMCS have similar relative density?

Missing validation procedure to verify the quality of the samples, GD&T

Did not see comparison graphics of simulation and experimental tests

Version 2:

Reviewer comments:

Reviewer #1

(Remarks to the Author)

The authors have satisfactorily solved most of reviewer's concerns. A remaining point that need to be addressed before publication:

Response (2) remains insufficient. The reviewer strongly suggest adding a demonstration figure with results that highlights the programmable functionalities and their application use cases, particularly in three-dimensional, real-world scenarios. This request is not tricky and should be addressed rigorously to broaden the manuscript's appeal to a general audience suitable for a high-impact venue such as Nature Communications, rather than a niche outlet focused on 2D mechanical energy absorption.

Reviewer #2

(Remarks to the Author)

The second round of review requested clarification regarding the comparison of the CMCS to the re-entrant metamaterial either through further results or extended analysis. The authors provide outstanding clarification of the SEA scaling with density via extension of their Ashby plot. In conjunction with the responses to Reviewer #1, I am thoroughly convinced of the authors' claims and recommend the paper for publication. I look forward to future investigations that consider specific applications of the CMCS.

REPLY

Ref: NCOMMS-25-27612

The authors sincerely thank the associate editor and reviewers for their careful reading and valuable suggestions, which have allowed us to clarify ambiguities and significantly enhance the manuscript's quality. We have responded to each reviewer's comment point-by-point below. The reviewers' comments are restated in italics, followed by our detailed responses. Major revisions are highlighted in red within the revised manuscript.

Reviewer #1

This work proposes a chiral multi-curved shell (CMCS) metamaterial that utilizes the folding mechanics of shells to achieve enhanced energy absorption and recovery properties. The general concept of the work is interesting, but substantial improvements are necessary before it can be considered for publication. The main concerns from the reviewer are as follows:

- (1) The proposed design closely resembles existing origami metamaterials. The authors need to clearly justify how their design differs from these existing structures to demonstrate the novelty of the study.*

Reply:

Thanks for your helpful comment. Although our proposed design bears visual resemblance to existing origami metamaterials, such as Kresling, Tachi-Miura, and Yoshimura patterns, it differs fundamentally in its construction method and key local structural features.

These origami metamaterial configurations are typically formed by folding planar sheets, resulting in flat fold surfaces. The crease lines exhibit extremely lower stiffness compared to the fold surfaces. As a result, under compressive loading, stress tends to concentrate near the folds, thereby limiting the overall energy absorption capacity.

However, our proposed CMCS metamaterial is derived from an in-plane concentric polygon that is swept and rotated along an out-of-plane direction, resulting in a construction method that is fundamentally distinct from that of traditional origami metamaterials. This construction method integrates the compression-torsion characteristic of Kresling origami metamaterials, while transforming the easily deformable “two flat surfaces + one crease” configuration into a continuous

curved surface, thereby significantly enhancing both load-bearing capacity and energy absorption performance.

Hence, the construction method and key local structural features of CMCS metamaterials are, in nature, distinct from that of traditional origami metamaterials, representing a novel structural concept.

The relevant explanation has been added to the revised manuscript. Please see Lines 80-84 on Page 3 of the main text.

(2) *The design presented is essentially an integration of multiple one-dimensional CMCS cylinders into two-dimensional panel-like functional components. However, the current version of the manuscript does not include any results demonstrating the potential for extending this structure into three-dimensional lattices or more complex functional systems. This requires significant justification.*

Reply:

Thanks for your valuable suggestions. Based on our experiments and numerical simulations, it is challenging to extend CMCS cylinders into three-dimensional (3D) lattices while preserving the deformation mode of individual cells.

As shown in Figure R1a, a CMCS metamaterial with $3 \times 3 \times 3$ lattice is printed in a cuboid arrangement. Under compression, the deformation behavior of unit cells within the 3D metamaterial differs from that of a single compressed unit cell. This is primarily due to the altered upper and lower boundary conditions of the unit cells within the 3D lattice, which significantly reduce the constraint stiffness. When the internal force generated by the compression-torsion and buckling deformation reaches a certain threshold, the contact boundaries of two adjacent cells become unstable, thereby altering the overall deformation mode of the metamaterial, as illustrated in Figure R1b. Even when the stiffness at the unit cell interfaces is enhanced by increasing the material modulus at the connections and reducing the overall slenderness ratio, as provided in Figure R1c, it remains difficult to achieve the desired deformation mode. In addition, as shown in Figure R1d, Fang et al. [Nature, 2025, 639: 639 – 645] recently also proposed a chiral cylinder composed of rods with high enthalpy or high internal forces (they only discussed the 2D lattice case of the chiral cylinder in experiment). They theoretically demonstrated that, under compression, a rod part of the

chiral cylinder generates a substantial force component along the compressed surface at its upper or lower boundary. Therefore, when forming a 3D lattice, it is difficult for these cylinders or our CMCS cylinders to design a suitable unit cell boundary with stiffness sufficiently large relative to the internal forces, such that it can be approximated as a sliding boundary.

Figure R1. Three-dimensional lattices of CMCS metamaterials. a. CMCS metamaterial with a $3 \times 3 \times 3$

cuboid lattice and its deformation modes at different displacements under quasi-static compression. **b.** Mises stress nephograms at different displacements during quasi-static compression. **c.** CMCS metamaterial with enhanced constraint stiffness and its deformation modes at different displacements under quasi-static compression. **d.** 2D lattice chiral metamaterial proposed by Fang et al. [Nature, 2025, 639: 639 – 645].

Hence, it is extremely challenging to extend high-enthalpy chiral cylinders into 3D lattices while preserving the deformation mode of individual cells, as changes in boundary conditions significantly affect the deformation behavior of chiral cylinders with high internal forces. Although this issue falls outside the scope of our study, it has been discussed in the revised manuscript. Please see Lines 210-216 on Page 8 of the main text as well as Supplementary Note 11 on Page 20 of the supplementary information.

Although CMCS cylinders cannot be extended into a 3D lattice while preserving the deformation mode of individual cells, their 2D lattice is generally more suitable for impact protection applications, as such systems typically require the protective layer to be as thin as possible. Hence, CMCS metamaterials holds significant potential for practical engineering applications.

(3) A critical aspect of metamaterials is their programmability. The paper lacks sufficient discussion on the programmability of the proposed design and its mechanical and functional performance. Additional results and a more systematic investigation are needed; otherwise, the design's applicability remains highly limited.

Reply:

Thanks for your insightful comments. The programmability of CMCS metamaterials have been discussed and analyzed in the revised manuscript. As shown in Figure 3 of the main text, the plateau force can be tuned to meet different protection requirements by simply adjusting the thickness t , while the overall shape of the energy absorption curve remains largely unchanged. Moreover, by varying the parameters n and θ , the positive and negative stiffness characteristics of the energy absorption curve can also be tailored. These results demonstrate that geometric parameter variation offers an effective means to achieve programmable behavior in CMCS metamaterials.

Figures R2 and R3 illustrate the programmability of CMCS metamaterials through parallel connection and matryoshka-like configurations, respectively. By connecting two metamaterials with

positive and negative stiffness in parallel, the resulting structure exhibits a flatter plateau and a quasi-zero stiffness characteristic, which is beneficial not only for energy absorption but also for vibration isolation. Additionally, the matryoshka-like design enables the superposition of the force–displacement curves of two microstructures while conserving internal space, offering an efficient solution for compact energy-absorbing systems.

Related discussions and results have been added in the revised manuscript. Please see Lines 25 and 164-169 on Pages 1 and 5 of the main text as well as Supplementary Note 9 on Pages 16-18 of the supplementary information.

Figure R2. Parallel programming method. a. Configurations of the two parallel CMCS metamaterials. **b.** Force-displacement curves of each individual metamaterial, where the blue dotted line represents the metamaterial with $n = 4$ and the green dashed line represents that with $n = 6$. **c.** Force-displacement curve of the combined parallel metamaterial, where the red solid line represents the actual result of the

parallel metamaterial and the dash-dotted line denotes the sum of the two individual curves in Figure R2b.

Figure R3. Matryoshka-like programming method. a. Configurations of the larger and smaller CMCS metamaterials. **b.** Force-displacement curves of individual metamaterial, where the blue dotted line represents the larger metamaterial and the green dashed line represents the smaller one. **c.** Force-

displacement curves of the combined matryoshka-like metamaterial, where the red solid line represents the actual result of the matryoshka-like metamaterial and the dash-dotted line denotes the sum of the two individual curves in Figure R3b.

(4) *There is extensive research on energy-absorbing metamaterials, but the literature review in this work is insufficient. Numerous studies have used strut-based or shell-based structures to engineer nonlinear behaviors in lattices, and these should be appropriately reviewed, including but not limited to:*

*Moestopo, W. P., Mateos, A. J., Fuller, R. M., Greer, J. R., & Portela, C. M. (2020). Pushing and pulling on ropes: hierarchical woven materials. *Advanced Science*, 7(20), 2001271.*

*Gao, Z., Ren, P., Wang, H., Tang, Z., Wu, Y., & Wang, H. (2024). Additive manufacture of ultrasoft bioinspired metamaterials. *International Journal of Machine Tools and Manufacture*, 195, 104101.*

*Wang, P., Yang, F., Zheng, B., Li, P., Wang, R., Li, Y., ... & Li, X. (2023). Breaking the tradeoffs between different mechanical properties in bioinspired hierarchical lattice metamaterials. *Advanced Functional Materials*, 33(45), 2305978.*

Reply:

Thank you for your valuable suggestions, and we apologize for the insufficient literature review in the original manuscript. In the revised version, we have incorporated additional studies on energy-absorbing metamaterial, including:

- “Energy absorption in metamaterials is typically achieved through elastic or plastic nonlinear deformation of their microstructures²²⁻²⁶, such as honeycomb, lattice, and shell configurations²⁷⁻³². Upon low- or medium-speed impact, these microstructures generally undergo bending, compressing, or stretching, thereby absorbing the impact energy³³⁻³⁵”. Please see Lines 39-42 on Page 2 of the main text and References 25, 26, and 30-32.
- “Although Snapp et al.⁵⁸ discovered a Palm microstructure characterized by a long and smooth plateau phase, its energy absorption relies on multilevel local buckling, which limits the efficient utilization of the material. This behavior is analogous to the energy absorption mechanism observed in foams⁵⁹ and is generally unfavorable for material reusability.” Please

see Lines 57-60 on Page 2 of the main text and References 58 and 59.

- “It is worth noting that energy absorption curves obtained from quasi-static compression tests are generally considered reliable proxies for evaluating low- and medium-speed impact resistance⁷²⁻⁷⁴. This is because they capture the dominant energy-absorption deformations, such as bending, compression, and stretching, that are normally rate-insensitive⁷⁵⁻⁷⁷.” Please see Lines 235-238 on Page 8 of the main text and References 72-74.

(5) *Is the proposed strategy confined to hexagonal channels? If so, the generality of the design is limited and must be significantly improved.*

Reply:

Thanks for your valuable comment. The proposed strategy is not confined to hexagonal channels and can be applied to any polygonal geometry. For example, as illustrated in the following Figure R4 or Figure 3b of the main text, from quadrilaterals to octagons, the corresponding CMCS structures can be generated by sweeping and rotating. In fact, the proposed strategy is extendable to polygons with an infinite number of sides, ultimately approaching a circular form.

Figure R4. Generated CMCS microstructures from quadrilaterals to octagons.

(6) *The reported properties lack systematic comparison with existing structures in the literature. Although Figure 1c provides some comparisons, they are limited and based solely on author-defined cases. A more comprehensive comparison, such as Ashby charts with a broader set of topologies, is essential to evaluate the true impact of this work.*

Reply:

Thanks for your helpful suggestions. Two Ashby plots have been included in the revised manuscript to compare the efficiency of energy absorption (EEA) and specific energy absorption (SEA) of CMCS metamaterials with those of other existing designs fabricated with TPU, as

provided in the following Figure R5. The results show that CMCS metamaterials exhibit significantly higher EEA across all density levels and achieve higher SEA than most other designs at comparable densities. Hence, CMCS metamaterials possess significant advantages in energy absorption performance. Please see Figures 4f and 4g on Page 7 and Lines 206-209 on Page 8 of the main text.

Figure R5. Ashby plots comparing the EEA and SEA of CMCS metamaterials with those of other existing metamaterials.

(7) *The authors must be very careful in discussing energy absorptions, as it can differ significantly across quasi-static, dynamic, and impact.*

Reply:

Thanks for your valuable suggestion. The force-displacement curve under quasi-static compression is suitable for evaluating the protective performance of structures under low- and medium-speed impacts. Such impacts typically involve large impulses and low loading rates, allowing sufficient time for the structure to deform and absorb energy through mechanisms such as bending, compressing, and stretching. As a result, the energy absorption characteristics observed in quasi-static tests closely reflect the structural response under these conditions.

However, in ultra-high-speed ballistic impacts, the impulse is small and the loading duration is extremely short, leading to a dominance of inertial effects, stress wave propagation, and rate-dependent failure modes such as fracture and shear perforation. These dynamic phenomena cannot be captured by quasi-static curves, making them unsuitable for assessing protective performance under high-velocity impact or penetration scenarios.

Hence, the energy absorption metrics obtained under quasi-static conditions are generally

applicable to low- and medium-speed impact scenarios but are not suitable for evaluating responses under ultra-high-speed impacts. Based on this point, some statements about quasi-static or impact energy absorption have been modified or replenished, which are as follows:

- “Upon impact, these microstructures undergo bending, compressing, or stretching, thereby absorbing the impact energy” has been revised to “Upon low- or medium-speed impact, these microstructures generally undergo bending, compressing, or stretching, thereby absorbing the impact energy.” Please see Lines 40-42 on Page 2 of the main text.
- “making it more suitable for impact or shock engineering” has been changed to “making it more suitable for impulse-driven impact or shock engineering”. Please see Line 89 on Page 3 of the main text.
- Explanatory content has been added to clarify the relationship between quasi-static and impact energy absorption, as follows: “It is worth noting that energy absorption curves obtained from quasi-static compression tests are generally considered reliable proxies for evaluating low- and medium-speed impact resistance⁷²⁻⁷⁴. This is because they capture the dominant energy-absorption deformations, such as bending, compressing, and stretching, that are normally rate-insensitive⁷⁵⁻⁷⁷. Therefore, the quasi-static curve provides a sound basis for assessing the energy absorption of metamaterials under low- to medium-speed impact loading conditions.” Please see Lines 235-239 on Page 8 of the main text.

Reviewer #2

The paper presents an intuitive, yet novel and noteworthy, design concept for energy absorbing mechanical metamaterials, called chiral multi-curved shells, that should be of interest to the applied mechanics community. Numerical and experimental data support the hypothesis that the design efficiently transfers axial loads to the distributed deformation of the metamaterial walls, thereby resisting compression of the entire body without localizing strain. This result is contrasted with the behavior of hexagon and Kresling metamaterials which, respectively, excessively localize strain and poorly transfer axial loads to wall deformations. A parametric study is performed to understand the role of the design architecture and optimize the structure according to metrics for energy absorption. The capabilities for applications in impact mitigation are then discussed. Sufficient details are provided to enable reproduction of the fabrication, simulation, testing, and data analysis presented.

The results appear significant and valuable, and the conclusions presented the "Chiral multi-curved shell (CMCS) metamaterials with compression-torsion and buckling mechanisms" and "Optimizing metamaterials for an excellent energy absorption curve" sections are logically sound with sufficient evidence. However, I have three major concerns that should be addressed with revision:

(1) Insufficient explanation of metrics: The metrics used for evaluation of the metamaterials (specific energy absorption, energy absorption efficiency, plateau phase, densification displacement, and undulation of load-carrying) are defined in Supplementary Note 5, rather than the main text or the methods, and are used without clarification of their significance. The text frequently refers to an ideal energy absorption behavior, which is presented in Supplementary Note 6, but this concept does not appear to be clarified in supporting citations 41 or 52. These details should be elucidated in the main text for the paper to reach the general audience of the journal rather than the audience of a technical journal devoted to energy absorption. Without further explanation or expertise, it is not possible to interpret the

significance of the results and to accept the conclusions of the paper.

Reply:

Thanks for your helpful suggestions. Definitions of densification displacement, plateau phase, undulation of load-carrying (ULC), efficiency of energy absorption (EEA), and specific energy absorption (SEA) have been added to the Methods section of the revised manuscript. Please see Lines 287-317 on Pages 10 and 11 of the main text. Further explanations on these metrics have been provided in the revised version, as detailed below:

- “A larger densification displacement D_D typically makes the normal working range longer, enabling greater energy absorption.” Please see Lines 296 and 297 on Page 10 of the main text.
- “A longer plateau displacement D_P allows for extended energy absorption before densification occurs.” Please see Lines 302 and 303 on Page 10 of the main text.
- “A smaller ULC χ_{ULC} generally results in a more gradual variation of the plateau force.” Please see Lines 306 and 307 on Page 11 of the main text.
- “A higher EEA implies that more energy can be absorbed before reaching the peak or damage threshold.” Please see Lines 311 and 312 on Page 11 of the main text.
- “An increased SEA corresponds to more energy absorbed per unit mass, reflecting enhanced material utilization.” Please see Lines 316 and 317 on Page 11 of the main text.

Additionally, an ideal energy absorption curve has also been elucidated in the Methods section of the revised manuscript. Please see Lines 318-325 and Figure 6 on Page 11 of the main text.

(2) *Unfair comparison of designs: In contrast with the thorough comparison of the CMCS to the hexagon and Kresling metamaterial, it is not evident the comparison of the CMCS to the re-entrant metamaterial is appropriately controlled. While the results presented are normalized with respect to the pressure area and the plateau force per unit area is matched via tuning of the thicknesses, there are multiple variables in the re-entrant metamaterial, particularly the number of cells, which are not discussed but may influence the results such as the specific energy absorption that depends on the total mass of the system. Furthermore, the pressure areas reported in Supplementary Note 12 should be properly motivated, specifically that of the CMCS metamaterial for which each tube has an open end but the reported pressure area is three times larger than that of the re-entrant metamaterial. Thus, the logic of the comparison used to reach*

the conclusion "It is evident that the CMCS metamaterial exhibits an excellent energy absorption curve and outstanding energy absorption performance" should be supported with additional evidence or logical arguments.

Reply:

Thanks for your valuable comments. We have added two comparisons, re-entrant metamaterial II and III, with different cells, as shown in Figure R6. For a fair comparison, namely ensuring their applicability in the same protective scenarios, we adjusted the thicknesses to achieve similar plateau forces per unit area across the four metamaterials. Furthermore, the three re-entrant metamaterials have the same mass per unit area. As shown in Figure R7 and Table R1, the energy absorption metrics of re-entrant metamaterial II and III do not differ qualitatively from those of the re-entrant metamaterial I, and the SEA shows no significant improvement. All values are lower than those of the CMCS metamaterial, with the SEA of the CMCS being nearly three times higher than those of the three re-entrant metamaterials. Additionally, two Ashby plots have been provided to compare the EEA and SEA of CMCS metamaterials with those of other metamaterials in references, as shown in Figure R8. It can be seen that CMCS metamaterials exhibit significantly higher EEA across all density levels and achieve higher SEA than most other designs at comparable densities. These results support the conclusion "It is evident that the CMCS metamaterial exhibits an improved energy absorption curve and enhanced energy absorption performance." Related comparisons and detailed information have been provided in the revised manuscript. Please see Figures 4f and 4g on Page 7 of the main text, Lines 206-209 and 231-233 on Page 8 of the main text, as well as Supplementary Note 12 on Pages 21-23 of the supplement information.

Figure R6. Configurations of the four metamaterials used for the comparison.

Figure R7. Comparison results of the four metamaterials. a. Force-displacement curves, where S is the pressure area. **b.** Radar chart displays EEA, SEA, plateau phase D_p , densification displacement D_d , and undulation of load-carrying (ULC) χ_{ULC} .

Table R1. EEA, SEA, plateau phase, densification displacement, and ULC values for the CMCS and the three re-entrant metamaterials.

Energy absorption metric	CMCS metamaterial	Re-entrant metamaterial I	Re-entrant metamaterial II	Re-entrant metamaterial III
EEA	0.88	0.74	0.72	0.74
SEA (J/kg)	200	64	66	66
D_P (mm)	0.352	0.288	0.289	0.327
D_D (mm)	0.463	0.342	0.346	0.372
χ_{ULC}	0.13	0.139	0.139	0.138

Figure R8. Ashby plots comparing the EEA and SEA of CMCS metamaterials with those of other existing metamaterials.

Considering practical engineering applications, metamaterials are typically used as core materials sandwiched between two rigid plates. Therefore, the equivalent pressure area is generally defined as the single-connected domain covered by the metamaterial, rather than the actual contact area. Hence, the pressure area of the CMCS metamaterial is the area controlled by periodic constants, namely $42 \text{ mm} \times 42 \text{ mm} \times 9$, which is nearly three-times larger than that of the re-entrant metamaterial I. Related explanations have been provided in the revised manuscript. Please see Lines 261-264 on Page 21 of the supplementary information.

Furthermore, the bottom area of the re-entrant metamaterial has no effect on the energy

absorption curve shown in Figure R7a, as the vertical axis represents the force normalized by the bottom area. Even if the bottom area of the re-entrant metamaterial is made equal to that of the CMCS metamaterial, the total force would increase proportionally according to the superposition principle. As a result, the normalized energy absorption curve in Figure R7a would remain unchanged. Hence, as long as the deformation mode remains unchanged, the bottom size of re-entrant metamaterials in the direction perpendicular to the periodic plane has little influence on the comparison results; using a smaller area can also improve research efficiency.

In addition, since the five metrics compared in Figure R7b are independent of area, their values remain unchanged regardless of how the pressure area is defined.

(3) *Missing connection from statics to impact: The drop tests compare force measurements in the presence of a CMCS and with no energy absorber. While this provides sufficient evidence to support the claims regarding the reusability of the metamaterial as fabricated with TPU, the claim "These results confirm the excellent impact protection performance..." requires additional evidence or logical arguments directly connecting the quasi-static testing presented in the rest of the paper to the dynamic drop-test for impact testing. Furthermore, any implications of changing the printed layer thickness from 0.16 mm to 0.08 mm and from a 3x3 arrangement (Fig 3a) to an apparent 2x2 arrangement (Fig 3d) should be clarified. This connection is crucial to reach the statements made in "Potential applications" which are all impact related.*

Reply:

Thanks for your valuable comments. The force-displacement curve obtained from quasi-static compression tests can reflect a metamaterial's impact resistance by revealing its deformation behavior and energy absorption capacity under low- and medium-speed loading [Baroutaji et al. *Thin-Walled Structures*, 2016, 98, 337-350; Hu et al. *Composites Part A: Applied Science and Manufacturing*, 2016, 90, 489-501; Uddin et al. *Scientific Reports*, 2025, 15(1), 2837]. Upon low- and medium-speed impacts, the dominant energy absorption deformations, such as bending, compressing, and stretching, are generally rate-insensitive [Ni et al. *Composite Structures*, 2023, 321, 117254; Xiao et al. *International Journal of Impact Engineering*, 2022, 169, 104333; Li et al. *Composite Structures*, 2023, 324, 117526]. Therefore, the quasi-static compression curve serves as a reliable basis for evaluating the energy absorption of metamaterials under low- to medium-speed

impact loading. These arguments have been provided in the revised manuscript. Please see Lines 235-239 on Page 8 of the main text.

In addition, considering the limited size of the hole through which the hammer passes in the drop hammer machine as shown in Figure R9, we proportionally reduced the microstructure size from 40 mm to 30 mm, excluding the thickness t , and changed the arrangement from 3×3 to 2×2 . This modification ensures that the hammer, with the metamaterial attached, can pass through the hole without obstruction. As the microstructure size was reduced, the printing layer height was adjusted to a smaller value to ensure fabrication accuracy. As illustrated in Figure R10, two microstructures with a height of 30 mm are printed using 0.16 mm and 0.08 mm printing layer heights, respectively. It is evident that the larger printing layer height results in lower resolution. In contrast, the smaller layer height, though requiring a longer printing time, produces a more refined structure. To ensure the printing accuracy of the small-sized metamaterial, the smaller printing layer height, namely 0.08 mm, was adopted during fabrication. Relevant explanation has been provided in the revised manuscript. Please see Lines 241-243 and 280 on Pages 8 and 10 of the main text, respectively.

Figure R9. The hole through which the hammer passes in the drop hammer machine.

Figure R10. CMCS microstructures with 0.16 mm (left) or 0.08 mm (right) printing layer height.

Reviewer #3

(1) *Seems like the adjectives used in the first two paragraphs are a little bit superlative, or could be somehow exaggerated (extraordinary, exceptional, impressive). Same in line 65, the use of outstanding seems too much. As a reference, this seems more appropriate: significantly improved energy absorption.*

Reply:

Thanks for your helpful suggestions. These exaggerated objectives in the first two paragraph and line 65 have been modified in the revised manuscript. For example, “with extraordinary properties” has been revised to “whose properties are exotic compared to those of natural materials”, “exceptional” has been replaced with “special”, “impressive” has been removed, and “outstanding” has been changed to “significantly improved”. Please see Lines 32, 33, 38, and 67 on Page 2 of the main text.

(2) *Line 70, eliminate word extensive, line 72, change word outstanding, and also if you are going to claim that, compared to what, and how does this perform with other research projects?*

Reply:

Thanks for your valuable comments. The word “extensive” has been eliminated in the revised

manuscript. Please see Line 72 on Page 2 of the main text.

The word “outstanding” has been replaced with “reliable”. In this study, drop tests with or without the CMCS metamaterial were compared. The application of the CMCS metamaterial resulted in an 77.7% reduction in peak force compared to the case without it. Therefore, as you suggested, “outstanding” may be overly strong, and “reliable” is used instead to more accurately reflect the performance. Please see Line 74 on Page 2 of the main text.

(3) line 73, novel may not be again a proper selection of adjectives, many other metamaterials shown in other published works show good energy absorption properties, so it is not really a new approach.

Reply:

Thanks for your helpful comment. The word “novel” has been changed to “new”. Although many other metamaterials show good energy absorption properties, this study proposed a new design concept, namely CMCS metamaterials, which also demonstrate effective energy absorption. Therefore, we have adopted “new” in place of “novel” to maintain a more objective tone. Please see Line 74 on Page 2 of the main text.

In addition, we have included two Ashby plots to compare the EEA and SEA of CMCS metamaterials with those of other metamaterials from the literature, as shown in Figure R11. The results clearly demonstrate that CMCS metamaterials offer significant advantages in both EEA and SEA at comparable densities. Please see Figures 4f and 4g on Page 7 of the main text and Lines 206-209 on Page 8 of the main text.

Figure R11. Ashby plots comparing the EEA and SEA of CMCS metamaterials with those of other existing metamaterials.

(4) *Intro or a subsequent section must include a more comprehensive research literature analysis.*

Reply:

Thanks for your valuable suggestion. In the revised manuscript, more researches have been included and analyzed, for example:

- “Energy absorption in metamaterials is typically achieved through elastic or plastic nonlinear deformation of their microstructures²²⁻²⁶, such as honeycomb, lattice, or shell configurations²⁷⁻³².” Please see Lines 39 and 40 on Page 2 of the main text, as well as References 25, 26 and 30-32.
- “Although Snapp et al.⁵⁸ discovered a Palm microstructure characterized by a long and smooth plateau phase, its energy absorption relies on multilevel local buckling and does not fully utilize the material. This behavior is analogous to the energy absorption mechanism observed in foams⁵⁹ and is generally unfavorable for material reusability.” Please see Lines 57-60 on Page 2 of the main text, as well as References 58 and 59.
- “It is worth noting that energy absorption curves obtained from quasi-static compression tests are generally considered reliable proxies for evaluating low- and medium-speed impact resistance⁷²⁻⁷⁴. This is because they capture the dominant energy-absorption deformations, such as bending, compressing, and stretching, that are normally rate-insensitive⁷⁵⁻⁷⁷. Therefore, the quasi-static curve provides a sound basis for assessing the energy absorption of metamaterials under low- to medium-speed impact loading conditions.” Please see Lines 235-239 on Page 8 of the main text, as well as References 72-77.
- Two Ashby plots shown in Figure R11 have been provided to compare CMCS metamaterials with other metamaterials in references. Please see Figures 4f and 4g on Page 7 of the main text and References 64-70.

(5) *Subsequent section, before results and conclusion, should be considered to include information regarding theoretical background, materials and methods. After finishing, it was noted that there is a methods section at the end, and supplementary information, which helped us understand a little bit better the design phase. It is suggested that some material should be included to understand the whole approach in a better way.*

Reply:

Thanks for your helpful suggestions. In consideration of the journal's formatting requirements, part of the theoretical background has been included in the Methods section. This primarily covers several key metrics used to evaluate the energy absorption performance of metamaterials, including densification displacement, plateau phase, undulation of load-carrying (ULC), efficiency of energy absorption (EEA), and specific energy absorption (SEA). Presenting these metrics in the main text helps readers better understand the energy absorption effects. Additionally, a description on the ideal energy absorption curve has also been incorporated into the Methods section. Please see Lines 287-325 on Pages 10-11 of the main text.

In addition, to clarify the material and fabrication method used, we have further emphasized in the revised manuscript that all metamaterials and microstructures in this study were fabricated using 3D printing technology with TPU material. Please see Lines 276 and 277 on Page 10 of the main text.

(6) *Difficult to follow through the text and with figures, when too many ideas are presented in one figure (applies for fig 1 and 2), even some sections required to move to supplementary notes, which do not help to have a proper flow of information*

Reply:

Thanks for your valuable comments. To improve the readability, previous Figure 1 has been divided into two parts in the revised manuscript. The first part presents the construction method of the CMCS metamaterial and validates its energy absorption performance. The second part primarily discusses the energy absorption mechanism of the CMCS metamaterial. Please see updated Figures 1 and 2 on Pages 3 and 5 of the main text, respectively.

The original Figure 2 has been split into Figures 3 and 4 in the revised version. Previous Supplementary Figure 8, which contains the force-displacement curves corresponding to the original Figures 2d-f, has been incorporated into the updated Figure 3. Figure 3 now presents the configurations used for the parameter analysis along with their corresponding force-displacement responses. Figure 4 provides the associated EEA and SEA values, further analyzes the effect of the stiffener, and identifies an optimized configuration. Furthermore, two Ashby plots have been included in Figure 4 to compare the EEA and SEA of CMCS metamaterials with those of other

existing metamaterials, thereby highlighting the improved energy absorption performance of the proposed design. Please see updated Figures 3 and 4 on Pages 6 and 7 of the main text, respectively.

(7) *Good explanation for the reasoning behind the design of CMCS, from a Kresling design.*

Reply:

Thanks for your affirmation. To better explain the proposed design, the revised manuscript provides an improved description of the construction process of the CMCS configuration, which is “The inspiration of its compression-torsion mechanism comes from Kresling origami structures which guide this unique deformation mode through staggered spatial creases⁶¹. Kresling origami configurations are typically created by folding planar sheets, whereas the CMCS metamaterial can only be constructed using a fundamentally different approach. It is generated by sweeping and rotating a concentric polygon along the vertical axis (Supplementary Note 1), replacing the characteristic “two flat surfaces + one crease” of Kresling patterns with a continuous curved surface. This curved geometry facilitates controlled buckling, thereby enhancing the energy absorption capability”. Please see Lines 78-84 on Page 3 of the main text.

(8) *Line 93, mentions that comparison samples have similar mass, and feature parameters, how close they are in terms of mass, relative density, that can allow to guarantee a fair comparison?*

Reply:

Thanks for your comment and sorry for our unclear description. In the previous version, the masses of the 3D-printed comparison samples were provided in Supplementary Note 3, where the hexagon, Kresling, and CMCS metamaterials were reported to weigh 7.6 g, 7.7 g, and 7.7 g, respectively. Therefore, their masses can be considered as effectively identical for comparative purposes. As shown in Supplementary Figure 3, all three samples share the same plane polygon and height, resulting in identical relative densities. Hence, the comparison among them is fair and meaningful.

To clarify the comparison, the sample masses have been included in the title of Figure 1, which is “the three metamaterials are designed with the same mass, and their actual 3D-printed masses are 7.6 g, 7.7 g, and 7.7 g, respectively”. The ambiguous term “similar” in original Line 93 has been revised to “same” since their design masses are identical, and “feature parameters” has been

replaced with “plane polygon and height”. Please see Lines 99-101 and 109 on Pages 3 and 4 of the main text, respectively.

(9) *Line 143, check use of excellent,*

Reply:

Thanks for your careful reading. In the revised manuscript, “excellent energy absorption curve” has been replaced with “enhanced energy absorption performance”. Please see Line 151 on Page 5 of the main text.

(10) *Line 144, dozens? Meaning? Could it mean 20 something, or 200 something? Figure 2 shows several parameters, but it is not clear if all the combinations were tested, a summary table of the parameters, combinations, replicas should be included. It is important to mention how many replicas and if a design of experiments methodology was used to guarantee statistical confidence. Also, it is not clear how all these changes of parameters result in significant differences in mass, relative density, that may mislead the interpretation of results.*

Reply:

Thanks for your valuable comments and sorry for unclear description. In previous Figure 2, twenty-six samples were tested, so “dozens” has been replaced with “twenty-six” in the revised manuscript. Please see Line 152 on Page 5 of the main text.

In these experiments, only one parameters varies while others remain constant with the following values: $t = 1$ mm, $\theta = 60^\circ$, $n = 6$, $h = 40$ mm, $a = 20$ mm, $b = 18$ mm, $c = 1$ mm, and $w = 5\sqrt{3}$ mm, where the circumcircle of the polygon remains unchanged when n varies. A summary table was provided to present all combinations. Please see Supplementary Tables 1 and 2 on Pages 12, 13, and 19 of the supplementary information.

Considering the experimental cost, we limited our investigation to the influence of three parameters, thickness t , polygon n , and rotation angle θ , on the force-displacement curve, using uniformly spaced values. Since there may be an EEA extremum between rotation angles of 60° and 75° , we selected more densely spaced experimental points within this interval.

These changes of parameters result in differences in mass, and these masses were presented in Supplementary Tables 1 and 2. In addition, the corresponding relative densities have been included

in these tables. Please see Supplementary Tables 1 and 2 on Pages 12, 13, and 19 of the supplementary information.

(11)*Good comparison in the subsection with respect to the reentrant.*

Reply:

Thanks for your encouraging comment. We have also conducted two additional experiments on re-entrant metamaterials with different unit cell sizes to further enhance the reliability of the comparison. Please see Supplementary Note 12 on Pages 21-23 of the supplementary information.

(12)*Section of potential applications should be moved to conclusions, no need for figures, unless they were tested somehow.*

Reply:

Thanks for your helpful suggestion. Section of potential applications and its figures have been relocated to the supplementary information. Please see Supplementary Note 14 on Page 26 of the supplementary information. Potential applications were discussed in conclusions, which is “paving a new way for protection engineering applications, such as buildings, armored vehicles, safety helmets, and drones”. Please see Lines 271 and 272 on Page 10 of the main text.

(13)*Too many repetitions of superlative adjectives, line 230, outstanding.*

Reply:

Thanks for your valuable comment. The word “outstanding” has been changed to “improved” in the revised manuscript. Please see Line 261 on Page 10 of the main text.

In addition, we have carefully checked the entire manuscript and replaced all the exaggerated adjectives to ensure a more objective and academic tone. Please see Lines 25, 29, 32, 33, 38, 72-74, 151, 205, 233, 234, 248, and 269 of the main text.

(14)*As a summary, it is a good work, needs some polishing in terms of format and structure, tests are good, but need some more details on the structure of the experiments, and more supporting figures, and detailed analysis in text.*

Reply:

We sincerely appreciate your positive evaluation and valuable suggestions. In response to your comments, we have revised both the content and structure of the manuscript to improve its readability and logical flow. As noted in our responses above, more details on the experiments have been provided in the revised manuscript, and more supporting figures as well as detailed analysis in text have been included.

REPLY

Ref: NCOMMS-25-27612A

We are deeply grateful to the associate editor and reviewers for their thorough evaluation and insightful feedback, which greatly helped us refine the manuscript and address potential ambiguities. Below, we provide detailed, point-by-point responses to each comment. For clarity, reviewer comments are presented in italics, followed by our corresponding replies. Substantial changes made in the revised manuscript are marked in red for easy reference.

AE

As you will see, while the reviewers collectively find that your revisions significantly improved and clarified the points your manuscript makes, reviewer #1 and #2 still raises important concerns that remain to be addressed. In particular, the mass-density normalization of how your metamaterials are compared to existing metamaterials requires attention. More generally, there should be a strong effort made to address reviewer #1 and their concerns surrounding new functionalities and unique observations which they feel could be made clearer.

Reply:

Thanks for your careful reading and helpful suggestions. Detailed, point-by-point responses to each of the reviewers' comments have been provided. In particular, we have conducted additional experiments to demonstrate that the SEA of CMCS metamaterials surpasses that of existing metamaterials, addressing Comment 3 of Reviewer 1. Furthermore, based on normalized relative density, CMCS metamaterials exhibit superior energy absorption performance compared to re-entrant metamaterials, as confirmed by the trend analysis from supplementary experiments (see the response to Comment 1 of Reviewer 2). Additionally, we further clarify that CMCS metamaterials offer a novel impact-protection function that combines lightweight design, high strength, and reusability, along with enhanced EEA and SEA properties as well as good programmability for energy absorption. These advantages stem from their unique compressing-rotating-buckling deformation mode, as discussed in our response to Comment 1 of Reviewer 1.

Reviewer #1

The review thanks the authors for the revision. Most of the concern has been addressed. however, critical points as follows need to be addressed before publication:

- (1) The explanation of the difference between the proposed work and existing studies is not convincing enough. Currently, it is more like a technical explanation, where all structures can have technical differences in stress distributions and mechanisms. What we are more interested is the new functionalities, properties, performance, and unique observations given, which are still absent in current submission.*

Reply:

Thanks for your valuable comments and sorry for our previous inadequate explanation.

Compared to existing studies, CMCS metamaterials demonstrate a novel impact-protection function combining lightweight design, high strength, and reusability. They exhibit superior energy absorption properties, including enhanced efficiency of energy absorption (EEA) and specific energy absorption (SEA). Furthermore, CMCS metamaterials offer good programmability for energy absorption, enabling tailored mechanical responses under diverse loading conditions. These advanced functionalities, superior mechanical properties, and good programmability fundamentally stem from the unique compressing-rotating-buckling deformation phenomenon intrinsic to the CMCS architecture, clearly distinguishing our proposed design from previously reported structures.

For the novel impact-protection function combining lightweight design, high strength, and reusability, the following Figure R1, which can be found in Figure 5 on Page 9 of the main text, provides the corresponding demonstration. Even though the CMCS metamaterial's relative density is only 40 % of that of the re-entrant metamaterial, it delivers comparable load-bearing capacity and absorbs more energy during the working phase, highlighting its lightweight design and high strength. After 20 consecutive drop-hammer tests, the CMCS metamaterial continued to deliver stable impact-protection performance, demonstrating its reusability.

Figure R1 Demonstration for the novel impact-protection function combining lightweight design, high strength, and reusability. **a.** Force-displacement curves of the CMCS (red line) and re-entrant (blue line) metamaterials by quasi-static compression experiments, where the relative densities of the CMCS and re-entrant metamaterials are 9.1% and 22.5%, respectively. **b.** The red areas represent the peak forces recorded over 20 drop tests with the CMCS metamaterial, and the black area denotes the peak force without it.

For the enhanced EEA and SEA properties, the updated Ashby plots of Figures 4f and g present the improvement, which are shown in the following Figure R2. Compared with existing metamaterials, the CMCS metamaterial exhibits an EEA increase of 0.08 over the highest previously reported value of 0.83, which can be regarded as a significant breakthrough [Snapp et al., Nature communications, 2024, 15(1), 4290]. Our additional experiments have confirmed that the SEA of CMCS metamaterials exceeds that of existing metamaterials fabricated with TPU, with an increase of 328 J/kg. A more detailed explanation of the enhanced EEA and SEA can be found in our response to your Comment 3.

Figure R2 Enhanced EEA and SEA properties. **a.** Ashby plot comparing the EEA of CMCS metamaterials with that of other metamaterials fabricated with TPU. **b.** Ashby plot comparing the SEA of CMCS metamaterials with that of other metamaterials fabricated with TPU.

For the good programmability of energy absorption, Figure 3 of the main text and Supplementary Note 9 discuss this capability. By adjusting geometric parameters or employing parallel/Matryoshka-like strategies, the force-displacement response can be tailored, enabling programmable energy absorption behavior in CMCS metamaterials.

In particular, in response to the earlier comment “*The proposed design closely resembles existing origami metamaterials. The authors need to clearly justify how their design differs from these existing structures to demonstrate the novelty of the study*”, we note that the impact-protection function of traditional origami metamaterials remains limited due to their reliance on flat panels and hinge-like crease lines, which result in inefficient material utilization. Some origami designs adopted multilayer configurations to absorb energy through multilevel local buckling, which is similar to the deformation behavior of foams, but these structures typically exhibit limited load-bearing capacity [Xiang et al., Materials & Design, 2021, 211, 110173]. Alternatively, incorporating specialized corner geometries can enhance load-bearing property [Zhai et al., PNAS, 2018, 115(9), 2032-2037]; however, this approach often introduces pronounced force peaks during compression, leading to a poor EEA [Wang et al., IJSS, 2024, 305, 113057]. Hence, origami metamaterials are inherently limited in improving their energy absorption performance due to the deformation modes determined by their architectures.

In contrast, the CMCS structure features intricately designed curved shells that promote multiple deformation modes during compression, thereby maximizing material utilization and markedly enhancing the aforementioned functionalities and properties at the same time. For instance, as shown in Figure 1c, the SEA of the CMCS metamaterial is twenty times higher than that of the Kresling origami counterpart, representing a substantial improvement in energy absorption.

Furthermore, a new deformation process can be observed in Figure 2 or Supplementary Video 1. In existing origami metamaterials, deformation typically initiates along the crease lines, while the panels remain largely unbuckled. By contrast, CMCS metamaterials buckle immediately under the synergy of compression-torsion and curved shell, enabling most of the material to engage in deformation and ensuring a gentle evolution of the structure’s geometric features without abrupt changes. This unique compressing-rotating-buckling deformation phenomenon yields a smooth force-displacement response with a high plateau phase, making CMCS metamaterials more closely approximate ideal energy absorption behavior.

In summary, compared with existing metamaterials, particularly origami-based designs, CMCS metamaterials offer the novel lightweight, high-strength, and reusable impact-protection function, enhanced EEA and SEA properties, and good programmability for energy absorption, all attributed to their unique compressing-rotating-buckling deformation mode.

Relevant explanations have been provided in the revised manuscript of the main text. Please see Lines 114, 115, 123-125, 140, and 141 on Page 4 of the main text.

(2) *The response “Based on our experiments and numerical simulations, it is challenging to extend CMCS cylinders into three-dimensional (3D) lattices while preserving the deformation mode of individual cells” is not acceptable. In fact, this can raise significant challenges since the proposed structures are shown to have limitations in 3D cases.*

Reply:

Thanks for your insightful comment and sorry for the oversight in our previous response.

Although the periodic 3D lattice metamaterial cannot perfectly preserve the deformation mode of individual units, the influence on its energy absorption performance is very limited. As shown in Figure R3, the experimental force-displacement curve of the 3D lattice remains relatively smooth during the plateau phase, closely resembling the anticipant curve assuming each unit maintains its ideal deformation mode. Meanwhile, the EEA reaches as high as 0.86, demonstrating that the structure maintains highly efficient energy absorption.

In addition, by employing a gradient design, each unit in the 3D-arranged CMCS metamaterial can preserve its ideal deformation mode, as illustrated in Figure R4. The gradient design enables multi-stage deformation and hierarchical energy absorption by progressively tuning the structural stiffness and plateau phase across layers, thereby broadening the effective amplitude range for impact protection [Zhao et al. Composite Structures, 2023, 321, 117312]. Moreover, by adopting a negative gradient design, a 'soft-inside-hard-outside' configuration can be achieved, which is more effective in reducing the impact peak and enhancing overall protection performance [Xiang et al. Thin-Walled Structures, 2020, 157, 106993].

Gradient arrangement	Layer	t (mm)	θ (deg)	w (mm)
	Top	1.0	90	$5\sqrt{3}$
	Middle	1.1	75	$6\sqrt{3}$
	Bottom	1.2	60	$7\sqrt{3}$

Hence, CMCS metamaterials can be extended into 3D lattices while maintaining reliable impact protection performance. In the revised manuscript, the previous descriptions have been revised in accordance with the above discussion. Please see Lines 217 and 218 on Page 8 of the main text and Supplementary Note 12 on Pages 21 and 22 of the supplementary information.

(3) *The results of the Ashby charts only show moderate advantages in SSA and SEA.*

Reply:

Thanks for your careful reading and sorry for our insufficient description.

The theoretical range of EEA ('SSA' is likely a typographical error and should be 'EEA') lies between 0 and 1, and further improvements become progressively more difficult as the value rises. For example, Snapp et al. [Nature communications, 2024, 15(1), 4290] proposed a metamaterial with an energy-absorbing efficiency K_s of 75.2%, a metric defined similarly to the EEA. Although this represents only a 3.4% improvement over the previous highest value of 71.8%, it was regarded as a major breakthrough. For our CMCS metamaterials, the EEA reaches 0.91, having an improvement of 0.08 (or 8%) over the highest value of 0.83 reported for other metamaterials in the Ashby plot shown in Figure 4f, which is a significant advancement. Additionally, for energy-absorbing structures or materials, lower density is generally preferred. In other words, the closer a data point lies to the upper left corner of Figure 4f, the better it is. Hence, although the absolute improvement in EEA is small, this incremental gain represents a meaningful step toward approaching the ideal value of 1.

Regarding SEA, since high-density CMCS metamaterials had not been previously tested, their advantage was not particularly prominent in absolute terms across all density levels. As shown in Figure 4a, increasing the thickness (and thus the mass or density) significantly improves the SEA. To further validate this, we have conducted additional experiments by increasing the thickness, as provided in Table R1, achieving higher SEA values. These results are presented in Figure 4g or the subsequent Figure R5. It can be observed that CMCS metamaterials exhibit an advantage in terms of SEA, with a maximum SEA exceeding that of other metamaterials by 328 J/kg, while maintaining relatively low density which is an essential attribute for impact protection applications. Hence, CMCS metamaterials outperform other metamaterials in SEA, combining lightweight design with high energy absorption.

Table R1 Additional experiments where h , b , c , and w are 40 mm, 18 mm, 1 mm, and $5\sqrt{3}$ mm, respectively.

Number	t (mm)	n	θ (deg)	Density (kg/m ³)	SEA (J/kg)
1	2.4	6	60	182.8	622
2	3.0	6	60	213.4	866
3	4.0	4	60	201.4	729
4	5.0	4	60	235.5	1000
5	6.0	4	60	267.0	1506
6	4.0	6	45	277.2	1661
7	5.0	6	60	303.6	1491
8	6.0	6	60	341.6	1835
9	4.0	7	60	271.1	1215
10	6.0	7	60	361.5	1825

Figure R5 Ashby plot comparing the SEA of CMCS metamaterials with that of other metamaterials fabricated with TPU.

Based on the above discussion, CMCS metamaterials demonstrate meaningful advancements in both EEA and SEA. Related discussions and experimental data have been included in the revised manuscript. Please see Figures 4f and g on Page 7 as well as Lines 212-216 on Page 8 of the main text, and Supplementary Note 11 on Page 20 of the supplementary information.

Reviewer #2

The initial review requested (1) elaboration of metrics used in analysis, (2) confirmation designs are fairly compared, and (3) more rigorous connection between static and impact analysis. The following revision mostly address the requests and significantly enhance the paper, but there remains one concern regarding the control used in Point (2):

(1) The metrics are clearly explained using both mathematical formulae and plain language in the Methods section. Furthermore, the plain language explanation of the metrics enable a broader audience to appreciate the physical significance of the ideal energy absorption curve and the consequences for deviations from the ideal curve.

(2) Additional re-entrant architectures with a variable number of cells and wall thicknesses, but fixed mass density, are tested. Ashby plots that showcase the scaling of the novel architecture in comparison to existing architectures are constructed.

(3) Satisfactory references are provided to rigorously connect the static response to the impact response.

The superiority of the CMCS to the re-entrant metamaterial is primarily quantified according to the SEA metric which depends on the force integrated up to the densification displacement and the total mass. Therefore, the conclusion that the CMCS is generally superior to the re-entrant metamaterial, rather than for specific cases, requires showing that the mass of the re-entrant metamaterial increases faster than the integrated force as features such as the number of cells or wall thickness are changed. The scaling of the re-entrant metamaterial SEA versus mass density on the Ashby plot (which may currently be labeled as auxetic?) would resolve this concern once and for all. If this follows directly from results that are reported in the existing paper, then it should be clarified.

Reply:

Thanks for your valuable comments and sorry for our unclear description. These points labeled as auxetic in the Ashby plot of Figure 4g are reported in the existing paper [Zou et al., Mechanics of Advanced Materials and Structures, 2024, 1-15], and are not experimental results obtained in our work. To clarify this content, a sentence has been provided in the revised manuscript, which is “where all SEA values except those of CMCS metamaterials are sourced from existing studies^{52,58,64-}

⁷⁰. Please see Lines 198 and 199 on Page 7 of the main text.

Furthermore, we have added the relative densities of the re-entrant metamaterial I, II, and III shown in Supplementary Figure 13, which are 22.5%, 22.4%, and 22.3%, respectively, and that of the CMCS metamaterial is 9.1%. Even though the densities of the re-entrant metamaterials are nearly 2.5 times higher than that of the CMCS metamaterial, their SEA values are only about one-third of that of the CMCS, further highlighting the superior energy absorption performance of the CMCS metamaterial. Please see Lines 290 and 291 on Page 23 of the supplementary information.

Moreover, several additional experiments on CMCS and re-entrant metamaterials with different thicknesses have been conducted to observe the trend of SEA as the mass density or relative density increases. Figure R6 compares the variation in SEA of CMCS and re-entrant metamaterials with increasing relative density. Although the SEA of both metamaterials increases with rising relative density, the growth trend of CMCS metamaterials is significantly steeper and consistently higher than that of the re-entrant metamaterials. In addition, at the same relative density of 28.7%, the SEA of the CMCS metamaterial is up to 16.4 times that of re-entrant metamaterials, representing a qualitative leap in energy absorption performance. Therefore, it is reasonable to infer that CMCS metamaterials are superior to re-entrant metamaterials in SEA. Please see Lines 325-340 on Page 27 of the supplementary information and Lines 235 and 236 on Page 8 of the main text.

Figure R6 Comparison of SEA trends for CMCS and re-entrant metamaterials with increasing relative density. For CMCS metamaterials, the thickness varies from 1.0 mm to 6.0 mm, with other geometric parameters consistent with those in Supplementary Figure 3. For re-entrant metamaterials, the thickness values are 0.308 mm, 0.35 mm, and 0.4 mm, with other geometric parameters identical to those of re-entrant metamaterial I shown in Supplementary Figure 13b.

Reviewer #3

(1) *Line 25 programmability, probably is better to use tuneability*

Reply:

Thanks for your helpful suggestion. This word “programmability” has been replaced with “tuneability”. Please see Line 25 on Page 1 of the main text.

(2) *Also from line 25. Compared to 25 traditional compression-torsion or buckling ones, the metamaterial achieves a 20-fold increase in 26 specific energy absorption (SEA) through a higher plateau force or a 50% increase in efficiency 27 of energy absorption (EEA) due to a gentler plateau phase, respectively. Compared to what, the metamaterial has either feature? Kind of confusing the sentence.*

Reply:

Thanks for your valuable comments and sorry for our unclear description. As illustrated in Figure 1b, the compression-torsion metamaterial corresponds to the Kresling pattern, whereas the buckling metamaterial is the hexagonal configuration. To clarify this content, this sentence has been revised to “Compared to a compression-torsion metamaterial with Kresling pattern and buckling metamaterial with hexagonal configuration, the proposed design achieves a 20-fold increase in specific energy absorption (SEA) through a higher plateau force and a 50% increase in efficiency of energy absorption (EEA) due to a gentler plateau phase, respectively”. Please see Lines 25-29 on Page 1 of the main text.

(3) *Line 32, exotic, change for non-conventional behavior or something like that.*

Reply:

Thanks for your valuable suggestion. “whose properties are exotic compared to those of natural materials” has been changed into “that exhibit non-conventional behavior not typically found in natural materials”. Please see Line 33 on Page 2 of the main text.

(4) *Line 44, recommended to introduce or explain what is a chiral microstructure.*

Reply:

Thanks for your helpful comment. An explanation on chiral microstructure has been provided in the revised manuscript, which is “whose mirror configuration cannot be superimposed into themselves”. Please see Lines 45 and 46 on Page 2 of the main text.

(5) *Line 67, careful with statements that make claims of better performance without reference of how the comparison is made. How much is significant improved? Better statement in line 70 of a clear comparison.*

Reply:

Thanks for your careful reading and valuable suggestion. The phrase “with significantly improved energy absorption” has been revised to “for energy absorption”. Please see Line 69 on Page 2 of the main text.

(6) *Again in line 74, a claim without much support and using a kind of ambiguous adjective (reliable)*

Reply:

Thanks for your helpful comment. “reliable impact protection” has been modified into “effective impact attenuation compared with the unprotected case”. Please see Line 76 on Page 2 of the main text.

(7) *For the experiments shown in fig, does the reentrant and the CMCS have similar relative density?*

Reply:

Thanks for your comments. The re-entrant and CMCS metamaterials do not share similar relative densities. For a fair comparison, we aim for both metamaterials to be applicable in the same protection scenario; therefore, the plateau forces of the two are similar, rather than their relative densities. Related explanation can be found in Lines 223-226 on Page 8 of the main text.

In addition, the relative densities of the re-entrant and CMCS metamaterials have been provided, which are 22.5 % and 9.1%, respectively. Please see Line 290 and 291 on Page 23 of the supplementary information.

(8) *Missing validation procedure to verify the quality of the samples, GD&T*

Reply:

Thanks for your careful reading and sorry for our oversight. In the revised manuscript, we have verified the quality of the 3D-printed samples of Figure 5b by measuring their mass and overall dimensions, as presented in Tables R2 and R3. It can be seen that the 3D-printed results closely match the expected values. Please see Supplementary Tables 4 and 5 on Page 25 of the supplementary information.

Table R2 Validating the quality of the 3D-printed CMCS metamaterial, where the measured value is shown on the left and the anticipated value on the right in each cell (measured | anticipated).

Sample	Mass (g)	h (mm)	a (mm)	b (mm)
CMCS metamaterial	68.6 69.0	39.99 40.00	20.01 20.00	18.02 18.00

Table R3 Validating the quality of the 3D-printed re-entrant metamaterial, where the measured value is shown on the left and the anticipated value on the right in each cell (measured | anticipated).

Sample	Mass (g)	Height (mm)	Length (mm)	Width (mm)
Re-entrant metamaterial	55.4 56.3	40.02 40.00	123.11 123.20	41.98 42.00

(9) *Did not see comparison graphics of simulation and experimental tests*

Reply:

Thanks for your valuable comment. The simulated stress nephograms of the CMCS and re-entrant metamaterials, along with comparisons to experimental tests, have been included in the revised manuscript. Please see Supplementary Videos 2 and 3. Some frames are shown in Figure R7, demonstrating that the experimental deformations closely match the simulated results.

Figure R7 Frames of Supplementary Videos 2 and 3.

REPLY

Reviewer #1

The authors have satisfactorily solved most of reviewer's concerns. A remaining point that need to be addressed before publication:

Response (2) remains insufficient. The reviewer strongly suggest adding a demonstration figure with results that highlights the programmable functionalities and their application use cases, particularly in three-dimensional, real-world scenarios. This request is not tricky and should be addressed rigorously to broaden the manuscript's appeal to a general audience suitable for a high-impact venue such as Nature Communications, rather than a niche outlet focused on 2D mechanical energy absorption.

Reply:

We thank for the insightful comment and it is very helpful to improve our manuscript. In response to your suggestion, we have revised a figure to highlight the programmable functionalities and their potential applications in three-dimensional, real-world scenarios, as shown in Figure R1. The high load-bearing capacity and reusability of the CMCS metamaterial make it a promising candidate for integration into aircraft landing gear cushioning structures. Furthermore, the programmability of the CMCS metamaterial enables the realization of graded plateau phases to accommodate different landing conditions, including normal landings, hard landings, and emergency reserves, thus achieving customizable functional performance. This figure and its related content have been provided in the revised version. Please see Supplementary Note 15 on Page 30 of the supplementary information.

[Figure Redacted]

Figure R1. Schematic illustration of the potential application of the CMCS metamaterial to aircraft landing-gear cushioning structures. a. Aircraft landing gear. **b.** Gradient design of the CMCS metamaterial. **c.** Experimental force-displacement curve, where m_{gra} is the mass of the gradient CMCS metamaterial and G is the gravity.

Reviewer #2

The second round of review requested clarification regarding the comparison of the CMCS to the re-entrant metamaterial either through further results or extended analysis. The authors provide outstanding clarification of the SEA scaling with density via extension of their Ashby plot. In conjunction with the responses to Reviewer #1, I am thoroughly convinced of the authors' claims and recommend the paper for publication. I look forward to future investigations that consider specific applications of the CMCS.

Reply:

We sincerely appreciate the reviewer's positive feedback and kind recommendation for publication. We are delighted that our extended analysis and clarification of the SEA scaling with density through the expanded Ashby plot addressed the reviewer's concerns regarding the comparison between CMCS and the re-entrant metamaterials. We are equally grateful for the encouragement to further explore application-oriented studies of CMCS metamaterials, which we view as a promising direction for our future research.